# The classical two-dimensional Heisenberg model revisited:
## An $SU(2)$-symmetric tensor network study

Philipp Schmoll[1,2*], Augustine Kshetrimayum[2,3], Jens Eisert[2,3], Román Orús[4,5,6], Matteo Rizzi[7,8]

**1** Institute of Physics, Johannes Gutenberg University, 55099 Mainz, Germany
**2** Dahlem Center for Complex Quantum Systems, Freie Universität Berlin, 14195 Berlin, Germany
**3** Helmholtz Center Berlin, 14109 Berlin, Germany
**4** Donostia International Physics Center, Paseo Manuel de Lardizabal 4, E-20018 San Sebastián, Spain
**5** Ikerbasque Foundation for Science, Maria Diaz de Haro 3, E-48013 Bilbao, Spain
**6** Multiverse Computing, Paseo de Miramón 170, E-20014 San Sebastián, Spain
**7** Forschungszentrum Jülich, Institute of Quantum Control, Peter Grünberg Institut (PGI-8), 52425 Jülich, Germany
**8** Institute for Theoretical Physics, University of Cologne, D-50937 Köln, Germany
* philipp.schmoll@gmx.de

June 28, 2021

## Abstract

The classical Heisenberg model in two spatial dimensions constitutes one of the most paradigmatic spin models, taking an important role in statistical and condensed matter physics to understand magnetism. Still, despite its paradigmatic character and the widely accepted ban of a (continuous) spontaneous symmetry breaking, controversies remain whether the model exhibits a phase transition at finite temperature. Importantly, the model can be interpreted as a lattice discretization of the $O(3)$ non-linear sigma model in $1 + 1$ dimensions, one of the simplest quantum field theories encompassing crucial features of celebrated higher-dimensional ones (like quantum chromodynamics in $3 + 1$ dimensions), namely the phenomenon of asymptotic freedom. This should also exclude finite-temperature transitions, but lattice effects might play a significant role in correcting the mainstream picture. In this work, we make use of state-of-the-art tensor network approaches, representing the classical partition function in the thermodynamic limit over a large range of temperatures, to comprehensively explore the correlation structure for Gibbs states. By implementing an $SU(2)$ symmetry in our two-dimensional tensor network contraction scheme, we are able to handle very large effective bond dimensions of the environment up to $\chi_E^{\mathrm{eff}} \sim 1500$, a feature that is crucial in detecting phase transitions. With decreasing temperatures, we find a rapidly diverging correlation length, whose behaviour is apparently compatible with the two main contradictory hypotheses known in the literature, namely a finite-$T$ transition and asymptotic freedom, though with a slight preference for the second.

# 1  Introduction

The two-dimensional classical Heisenberg model has moved into the focus of attention as early as in the 1960ies [1,2] as a paradigmatic model for understanding magnetism in spin systems. The presumably most striking result achieved at that time was the Mermin-Wagner theorem: In systems featuring continuous symmetries in one and two spatial dimensions, there cannot be an instance of spontaneous symmetry breaking at any non-zero temperature [3]. As such, this statement applies to the classical Heisenberg model, too. Interestingly, at the same time, work was performed that seemed instead to indicate the existence of a phase transition at finite temperature [4, 5], building on high temperature expansions and spin-wave arguments. This apparent contradiction naturally sparked a significant interest in the community, with different authors employing different techniques to answer this question. This includes studies using Monte Carlo techniques [6–11], high temperature expansions [12,13], and other methods with contradictory results [14, 15]. Meanwhile, two other important aspects of the puzzle emerged, both with profound implications on our modern understanding of field theories, including quantum ones. The first is the fact that the absence of spontaneous symmetry breaking strictly speaking does not imply the absence of quasi-long-ranged order, and so-called *Berezinskii-Kosterlitz-Thouless* (BKT) transitions [16,17] may occur. Some studies have indeed found signatures resembling those of a BKT phase transition at $T_c \sim 0.6$ [18]. More recent works, using accurate finite-size scaling, seemed rather to suggest a pseudo-critical region [19] and quasi-long-range order [20]. The second one relies on the mapping between $D$-dimensional classical models and $(d+1)$-dimensional quantum

ones, with $D = d + 1$ [21–23]. The classical two-dimensional Heisenberg model should then share properties with the quantum field theory known as the $O(3)$ non-linear sigma model [21–23], with a bare coupling directly proportional to the classical temperature (see Section 2 for more details). The most striking feature of the latter is the non-perturbative phenomenon of asymptotic freedom [24, 25]: Namely the effective coupling constant being vanishing at large energies (short distances), while tending to get stronger and stronger at low energies (large distances), similarly to what happens in the celebrated case of $3 + 1$-dimensional *quantum chromodynamics* (QCD) [26–28]. According to this line of thinking, there should then be no doubt about the absence of whatsoever phase transition at finite temperature, with the essential physics being captured by an exponentially diverging correlation length while approaching $T = 0$ (again, see Section 2). However, such mappings to quantum field theories are strictly valid only in the continuum limit, with corrections originating from a finite lattice spacing possibly giving rise to additional effects, which in turn could be considered misleading or interesting features [29]. All in all, it is fair to say that the specifics of the phase transition of the classical Heisenberg model in two spatial dimensions has remained inconclusive despite all these attempts. This state of affairs is aggravated by the central role this model plays in quantum statistical mechanics. In this work, we provide a fresh perspective to this long-standing question, by making use of state-of-the-art *tensor network* (TN) [30–32] approaches to address the problem at hand. Specifically, we express the partition function of the classical Heisenberg model in two dimensions as an infinite tensor network consisting of $O(3)$-symmetric rank-four tensors for a given inverse temperature $\beta > 0$. We achieve such a formally exact rewriting via an expansion over the fundamental characters of the (non-Abelian) group, in a similar fashion to what was recently performed for the Abelian $O(2)$ case in Refs. [33, 34]. We then compute a number of relevant quantities such as spin-spin correlators, the thermodynamic entropy $S$ and geometric entanglement $\epsilon$, among others. This requires contracting the whole infinite two-dimensional tensor network. We employ a symmetry-preserving *corner transfer matrix renormalization group* (CTMRG) approach [35–38] in order to achieve this, similarly to what has been done for the quantum case of infinite *projected entangled pair states* (iPEPS) [39]. Our techniques have the added value of directly tackling the thermodynamic limit: Residual finite-size effects are encompassed by the so-called bond dimension of the ansatz and could be accounted for in a rather systematic way [40–44]. These features have made two-dimensional tensor networks a very suitable tool for studying intricate condensed matter problems, not only via their ground states [45–48] but even beyond [49–54], as well as finite temperature properties of both classical and quantum models in two spatial dimensions [33, 34, 55–62]. The present work builds upon and develops this substantial technical machinery. Taking profit from an $SU(2)$-invariant framework for tensor manipulations [63], we are able to achieve very large bond dimensions of the environment and thus to extrapolate our predictions in a presumably unprecedented way. To the best of our knowledge, this is the first study of the classical Heisenberg model in two dimensions using tensor network approaches and thus helps to shed further light onto the debated questions. The stable and efficient numerical machinery that we provide here also opens up a lot of exciting new opportunities to explore related models and materials, including ones in the realm of classical statistical mechanics.

This work is organized as follows. In Section 2, we define the model and we revise its diverse interpretations sketched above. In Section 3 we introduce the formalism to write the partition function as a tensor network. Following this construction, Section 3.3 describes how the infinite two-dimensional network can be contracted using a *corner transfer matrix* (CTM) scheme. Hence we not only compute fundamental thermodynamic quantities as the free energy and the entropy per lattice site (Section 3.4), but we also glance at quantum-inspired quantities such as the corner entropy and the geometric entanglement (Section 3.5) which could possibly highlight phase transitions. Moreover, as explained in

Section 4, we get efficient access to a number of observables, both local and at a distance. In particular, we discuss the bond energy as well as spin-spin correlation functions, that unveil the characteristic behaviour of the Heisenberg model at different temperatures. Finally, our work is concluded in Section 5 with some discussions and an outlook on possible future applications.

## 2 The model

The classical Heisenberg model represents the workhorse of statistical mechanics for capturing (anti-)ferromagnetism when no lattice distortion or any other anisotropy source plays a major role. Consequently, the constituent objects are three-dimensional vectors $\hat{S}_k$ of unit length, $|\hat{S}_k|^2 = 1$, living on the sites $k \in V$ and interacting along the links $\langle i, j \rangle \in E$ of a graph $G = (V, E)$ capturing the lattice structure of the system. The defining Hamiltonian then reads

$$H = -\sum_{\langle i,j \rangle} \hat{S}_i \cdot \hat{S}_j \,, \tag{1}$$

evidently invariant under actions of the $O(3)$ symmetry group, i.e., rotations and reflections. The central object for the study of statistical mechanics models is the partition function $\mathcal{Z}$, from which thermodynamic quantities, as well as local expectation values and correlation functions can be computed. At inverse temperature $\beta = 1/T > 0$ (setting $k_{\mathrm{B}} = 1$), it reads

$$\mathcal{Z} := \mathrm{Tr}\, \mathrm{e}^{-\beta H} = \int \mathcal{D}\Omega\, \mathrm{e}^{\beta \sum_{\langle i,j \rangle} \hat{S}_i \cdot \hat{S}_j} = \int \left( \prod_{k \in V} \frac{\mathrm{d}\Omega_k}{4\pi} \right) \left( \prod_{\langle i,j \rangle \in E} \mathrm{e}^{\beta \hat{S}_i \cdot \hat{S}_j} \right), \tag{2}$$

where $\mathrm{d}\Omega_k := \mathrm{d}\theta_k\, \mathrm{d}\phi_k \sin\theta_k$ is the infinitesimal surface element of the unit sphere at site $k$. We focus here on the apparently simple, yet very rich, square lattice in two spatial dimensions for the reasons we illustrated in the introduction and we deepen below in this section. Moreover, we deal here with the *ferromagnetic version* (FM) of the model, though we could equivalently work with the *anti-ferromagnetic version* (AFM) instead, since in the classical case the two models can be mapped into each other by a suitable transformation on bipartite lattices. We stress, however, that the entire tensor network construction of Section 3 is generally valid for any graph $G$ and interaction sign, a property which discloses the path for future studies in different scenarios, e.g., frustrated lattices.

Before dwelling on the details of our tensor network approach, let us recollect an important strategy to uncover the physics of the model at hand, namely through the investigation of an associated field theory. Condensed matter theorists are indeed usually interested in the universal properties arising in the neighbourhood of a phase transition. There, the correlation length $\xi$ diverges to values much larger than the lattice spacing $a$, or otherwise said the typical energy scale of excitations (gap) $\Delta$ vanishes with respect to any bare energy scale. Microscopic lattice details should not matter anymore: Correlation and response functions can be then extracted from a field theory defined in the continuum without intrinsic cutoffs (the lattice instead gives an ultraviolet momentum cutoff $\Lambda \simeq 1/a$). The exponent involved in the partition function is then interpreted as the action $\mathcal{S}$ of the theory, with one of the original dimensions playing the role of the (imaginary) time $\tau$, along which (Wick-rotated) Feynman path integrals are performed [21–23]. The most standard of these procedures consists in replacing the original $\hat{S}_k$ unit vectors with some $N$-component coarse-grained field $(\vec{x}, \tau) \mapsto \vec{\phi}(\vec{x}, \tau)$ and then getting out a $\phi^4$ Ginzburg-Landau-Wilson functional: This is the realm to which the machinery of symmetry-breaking mechanisms or its forbiddance via the Hohenberg-Mermin-Wagner

theorem finds application. However, an even more suggestive possibility – making use of vector fields $(\vec{x}, \tau) \mapsto \vec{n}(\vec{x}, \tau)$ which preserve the essential property of having unit length – is offered by the $O(N)$ *non-linear sigma model* (NLSM) [21–23]

$$\mathcal{S}_n = \frac{N}{2cg} \left[ (\partial_\tau \vec{n})^2 + c^2 (\partial_x \vec{n})^2 \right] , \tag{3}$$

with the spin-wave velocity set to unity, $c = 1$, and the bare coupling $g = Na^{d-1}/\beta$ reading simply $g = 3/\beta$ for the $N = 3$, $d := D - 1 = 1$ case of our interest.

This simply formulated model shares a wealth of peculiarities with more intricate higher-dimensional ones, prominently with quantum chromodynamics (QCD) in $d = 3$. There, particle physicists want to make use of the lattice only as a convenient tool to investigate the field theory per se, i.e., they want to send the UV cutoff $\Lambda$ and the energy scale $J$ to infinite, while keeping all other scales ($\Delta$, $\xi$, etc.) fixed, which is usually achieved via renormalization group (RG) procedures. Along the decades, results first obtained in the large-$N$ limit via formally exact saddle-point expansions of the action have been refined to all $N \geq 3$.[1] The arguably most striking one is the so-called RG beta-function [21, 24, 25]

$$\frac{\mathrm{d}g}{\mathrm{d}\ell} = \frac{(N-2)}{2\pi N} g^2 + O(g^3) , \tag{4}$$

which unveils the cornerstone phenomenon of *asymptotic freedom*. The meaning thereof becomes apparent when we integrate the RG beta-function to predict what happens at a given scale of energy, or in this framework of (quantum) temperature $T_\mathrm{q} > 0$,

$$g(T_\mathrm{q}/c) \simeq \frac{g(\Lambda)}{1 + g(\Lambda) \ln(T_\mathrm{q}/c\Lambda) 2\pi N/(N-2)} , \tag{5}$$

where $g(\Lambda = 1/a)$ is the bare physical coupling we defined above at lattice scale. For $T_\mathrm{q} \to \infty$, i.e., at large energies and for short distances, the effective coupling vanishes and the theory is almost free, while for $T_\mathrm{q} \to 0$, i.e., low energies and large distances, the effective coupling gets stronger and stronger. This means that the physics resembles more and more the one of lower and lower $\beta$ values, i.e., the system is always magnetically disordered. Correspondingly, a sizeable gap for excitations develops, a behaviour often called "dynamical mass generation" and quite simplistically sketched as $\Delta_+ = c \Lambda \exp\left(-2\pi N/((N-2)g) + O(g^0)\right)$, which however holds only asymptotically and is therefore misleading in trying to fit numerical data. Summarising, the field theory approach strongly indicates a divergent correlation length $\xi = c/\Delta_+$ at low classical temperatures (i.e., high $\beta$ and low bare $g$ values), without any signature of a phase transition at finite values of it. However, all the mappings conducted above are strictly speaking valid only under the assumption of being close enough to the continuum limit (i.e., to the critical regime in classical terms), and lattice effects might give rise to additional effects. The latter are either treated as spurious by particle physicists or as interesting features by condensed matter ones [29]. Some observations are still in order before concluding this section and moving to our tensor network study and its results. The first one is that, by introducing the angular momentum operators $\vec{L}_k$ as the canonical conjugate to the position of a particle on the unit sphere, we could establish a mapping with the so-called $O(3)$ quantum rotor model, too,

$$H_R = +\tilde{g}J \sum_k \vec{L}_k^2 - J \sum_{\langle i,j \rangle} \vec{S}_i \cdot \vec{S}_j \qquad Z_R = \mathrm{Tr}\left(\mathrm{e}^{-H_R/T_\mathrm{q}}\right) , \tag{6}$$

---

[1]We recall here that the cases $N = 1$ (Ising) and $N = 2$ (XY) are substantially different in that they exhibit a phase transition of symmetry-breaking and topological (BKT) nature, respectively.

where the bare couplings read $\tilde{g} = 1/\beta^2$ and $J = \beta/a$. We also introduced a quantum temperature $T_q$ which corresponds to the finite size of the system along the (imaginary) time direction in the previous two models. The second one is that an analogue construction could be started from a purely quantum Heisenberg model in $d = 1$, where the $S_k$ are now quantum spin operators, leading at first sight to the same result of an $O(3)$-NLSM field theory. However, along the path we should be particularly careful in at least two respects. First, in noticing that at quantum level the ferromagnetic case is pretty different from the anti-ferromagnetic one, namely it has a different dynamical critical exponent, $z = 2$, thus impairing a straightforward $D = d + 1$ classical analogy. Second, we should account for geometric phases (a.k.a., Berry phases and their generalizations) which could arise through the rotating spins and contribute to the action with a topological term

$$\mathcal{S}_B = i\frac{\theta}{4\pi} \int \mathrm{d}x \int \mathrm{d}\tau \, \vec{n} \cdot (\partial_x \vec{n} \times \partial_\tau \vec{n}) = i\theta \mathcal{Q}, \tag{7}$$

with $\theta = 2\pi S$ and $\mathcal{Q}$ the so-called Pontryagin index. This is a topological invariant characterizing the configurations of $\vec{n}(\vec{x}, \tau)$ in spacetime, i.e., the homotopy classes via $\pi_2(S_2) = \mathbb{Z}$. Such a $\theta$-term (often also dubbed Weiss-Zumino action term) is the one responsible for the peculiarly different behaviour, gapped vs. gapless, of integer vs. half-integer spin Heisenberg chains in $d = 1$, at the heart of the celebrated Haldane conjecture [64]. Noticeably enough, such a topological term is also the one generated in the semi-classical context by certain lattice distortions and plays a central role in the description of Skyrmions (i.e., the defects with $|\mathcal{Q}| = 1$) in real materials [65–68]. Vice versa, in the absence of an explicit coupling, there exist quite compelling arguments to rule out such Skyrmions from playing the role of vortices in $O(2)$ theories, i.e., being responsible for topological transitions à la BKT [69].

## 3 The partition function as a tensor network

Here, we aim to use tensor networks as the numerical technique to compute the classical partition function. Tensor networks have been developed in the context of strongly-correlated quantum systems to represent many-body wave functions based on their entanglement structure. However, their network setup resembling the underlying physical lattice makes them equally well-suited for the simulations of classical models, by means of their partition function. In the following derivation, we will introduce a graphical convention that will conveniently lead to the tensor network structure of the partition function, resembling the underlying graph $G = (V, E)$ the model is defined on. We here substantially contribute to the study of classical spin models by means of tensor networks, but remark that the earliest historical studies, that can be identified as being tensor network studies for systems in two spatial dimensions, have been dedicated to the study of classical statistical models [70]. In some way, therefore, we build on this mindset and tradition here. By employing the character expansion of the rotational group $O(3)$, also known as the Rayleigh equation or plane-wave expansion in the contexts of electromagnetism and scattering theory, we can rewrite the Boltzmann factors on the edges $E$ of the graph in terms of a linear combination of products of terms living on the vertices $V$ according to

$$\mathrm{e}^{\beta \hat{S}_i \cdot \hat{S}_j} = \frac{(2\pi)^{3/2}}{\sqrt{\beta}} \sum_{\ell=0}^{+\infty} I_{\ell+\frac{1}{2}}(\beta) \sum_{m=-\ell}^{+\ell} Y_{\ell,m}^*(\hat{S}_i) \, Y_{\ell,m}(\hat{S}_j) \tag{8}$$

Here $\beta \mapsto I_{\ell+1/2}(\beta)$ are modified Bessel function of the first kind and $(\theta, \phi) \mapsto Y_{\ell,m}(\theta, \phi)$ are the spherical harmonics. In Eq. (8), we also introduce the graphical notation for the Boltzmann factors, from which the full partition function can be graphically derived. The Boltzmann factors on the links $E$ are drawn as an ellipse, which can be decomposed into a discrete sum of (products of) coefficients living on the vertices $V$. This rewriting, hence, transforms the integration over all site-variables $(\theta_k, \phi_k)$ in Eq. (2) into a product of independent integrations of all spherical harmonics living there. The explicit sum over $\ell$ and $m$ is denoted by the line connecting the two lattice sites, for which we can define the following two objects

$$
\begin{aligned}
\circ\!\!-\!\!\blacktriangleright \ (\ell, m) &= \left( \frac{(2\pi)^{3/2}}{\sqrt{\beta}} I_{\ell+\frac{1}{2}}(\beta) \right)^{1/2} Y_{\ell,m}^*(\hat{S}_i) \\
(\ell, m) \ \blacktriangleright\!\!-\!\!\circ &= \left( \frac{(2\pi)^{3/2}}{\sqrt{\beta}} I_{\ell+\frac{1}{2}}(\beta) \right)^{1/2} Y_{\ell,m}(\hat{S}_j)
\end{aligned}
\tag{9}
$$

The arrow on the edges is introduced in preparation for the symmetric tensor network ansatz we are going to develop. Note that the spirit of the mapping in Eq. (8) has also been used in previous works on the $O(2)$ XY model [33, 34]. The same decomposition can also be used in the anti-ferromagnetic version of the Heisenberg model, in which additional factors of $(i)^{2\ell}$ appear in the plane-wave expansion of Eq. (8). The missing parts for the construction of the partition function in Eq. (2) are the integrations of the two continuous parameters $(\theta_k, \phi_k)$ on all vertices $V$ of the lattice. In our graphical notation they will be denoted as

$$
\int \frac{\mathrm{d}\Omega_k}{4\pi} = \bigcirc.
\tag{10}
$$

Once a concrete graph for the lattice underlying the Heisenberg model is considered, the two parts of the graphical notation in Eq. (9) and Eq. (10) will directly lead to the tensor network structure, as we will demonstrate in the following sections.

## 3.1 Application to the one-dimensional Heisenberg chain

Let us start with the construction of the partition function for the classical Heisenberg spin-chain. This allows us to employ the graphical notation introduced above in a simple setting. Moreover, it will build an intuition for the tensor network appearing and the benefit of exploiting $SU(2)$ symmetry therein. Assuming a linear chain of three-dimensional spins $\hat{S}_k$, the full partition function becomes

$$
\begin{aligned}
\mathcal{Z} &= \cdots \ \langle\text{graphical network}\rangle \ \cdots \\
&= \cdots \ \langle\text{graphical network}\rangle \ \cdots
\end{aligned}
\tag{11}
$$

Exploiting the decomposition of the Boltzmann factors in Eq. (8), the spherical harmonics can be assigned to the vertices of the corresponding spins they act upon. The integration over the continuous variables can be performed locally, i.e., on every vertex independent of all the others. The partition

function in Eq. (11) is then simply the product of objects

$$(\ell_1, m_1) \longrightarrow\!\!\!\text{OO}\!\!\!\longrightarrow (\ell_2, m_2) \quad = \quad \prod_{i=1}^{2} \left( \sqrt{\frac{\pi}{2\beta}} I_{\ell_i + \frac{1}{2}}(\beta) \right)^{1/2} \\ \times \int \mathrm{d}\Omega_k \, Y_{\ell_1, m_1}(\theta_k, \phi_k) \, Y^*_{\ell_2, m_2}(\theta_k, \phi_k)$$

(12)

which live on the vertices of the lattice. This unveils a fundamental property of the construction, namely that the local objects can be decomposed into two parts: i) A so-called *degeneracy part* that includes numerical factors stemming from the Boltzmann factors on the edges $E$, and depends on the temperature and on the angular momenta principal numbers $\ell_j$ only; ii) a *structural part* that ensures that the angular momenta on the edges attached to each vertex are compatible. In particular, using the orthogonality of the spherical harmonics, we find

$$\int_{\theta=0}^{\pi} \int_{\phi=0}^{2\pi} \mathrm{d}\Omega_k \, Y_{\ell_1, m_1}(\theta_k, \phi_k) \, Y^*_{\ell_2, m_2}(\theta_k, \phi_k) = \delta_{\ell_1, \ell_2} \delta_{m_1, m_2}$$

(13)

for the two-edges vertices of a linear chain. For lattices with higher connectivity (i.e., a larger number of edges per vertex), the structural part will be *Clebsch-Gordan coefficients* (and combinations thereof) that dictate the coupling of spins, as we will demonstrate in the next section. At this point we can introduce the $SU(2)$-symmetric tensor network notation, in which the local objects in Eq. (12) will be denoted by a two-index tensor according to [63]

$$(\ell_1, m_1) \longrightarrow\!\!\bigcirc\!\!\longrightarrow (\ell_2, m_2) \quad \equiv \quad (\ell_1, m_1) \longrightarrow\!\!\!\text{OO}\!\!\!\longrightarrow (\ell_2, m_2)$$

$$= \quad (\ell_1) \longrightarrow\!\!\bigcirc\!\!\longrightarrow (\ell_2) \quad \times \quad (\ell_1, m_1) \longrightarrow\!\!\bullet\!\!\longrightarrow (\ell_2, m_2) \overset{\displaystyle \delta_{\ell_1, \ell_2} \delta_{m_1, m_2}}{}$$

(14)

with the *degeneracy part* being

$$(\ell_1) \longrightarrow\!\!\bigcirc\!\!\longrightarrow (\ell_2) \quad = \quad \prod_{i=1}^{2} \left( \sqrt{\frac{\pi}{2\beta}} I_{\ell_i + \frac{1}{2}}(\beta) \right)^{1/2} ,$$

(15)

and the *structural part* imposing the conservation of the angular momentum, $\ell_1 = \ell_2$, see Eq. (13). The evaluation of the partition function in Eq. (11) is now straightforward, since the angular momenta have to be the same across all links of the one-dimensional chain. Therefore, it reduces to a sum of powers of Bessel functions

$$\mathcal{Z}(\beta) = \sum_{\ell=0}^{\infty} \sum_{m=-\ell}^{+\ell} \left( \prod_{l \in E} \sqrt{\frac{\pi}{2\beta}} I_{\ell + \frac{1}{2}}(\beta) \right) = \left( \frac{\pi}{2\beta} \right)^{N/2} \sum_{\ell=0}^{\infty} (2\ell + 1) \left( I_{\ell + 1/2}(\beta) \right)^N ,$$

(16)

as it is also known from its exact solution (specifically, the model is Bethe integrable) [71, 72]. Here, $N$ denotes the number of edges of the chain. The factor $(2\ell + 1)$ is due to the degeneracy of the angular momentum eigenvalues $\ell$. Considering a chain with open boundary conditions in the thermodynamic limit, only the largest eigenvalue $\ell = 0$ contributes to Eq. (16) and the partition function becomes

$$\mathcal{Z}_N(\beta) \overset{N \to \infty}{\sim} \left[ \frac{\sinh(\beta)}{\beta} \right]^N ,$$

(17)

which is equivalent to Fisher's expression in Ref. [71]. In Section 3.4.1, we will see what this expression does tell us about thermodynamic quantities. Before doing so, we first proceed with the tensor network notation for the targeted square lattice.

## 3.2   Application to the two-dimensional square lattice

Making use of the graphical notation, the partition function for the two-dimensional Heisenberg model on a square lattice is given by

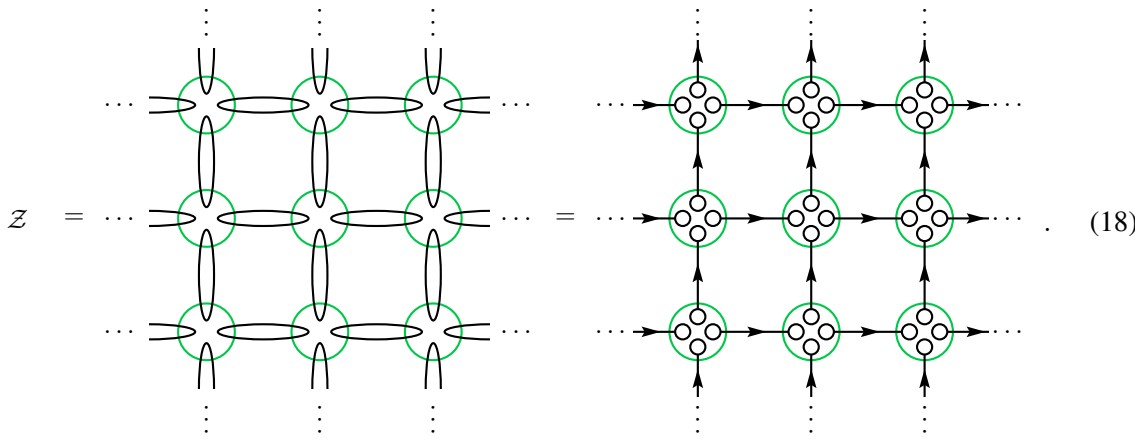

$$\mathcal{Z} \quad = \quad \cdots \qquad \qquad \cdots \quad = \quad \cdots \qquad \qquad \cdots \quad . \tag{18}$$

Similarly to the construction of the one-dimensional chain, the decomposition of the Boltzmann factors allows us to collect all terms into local objects

$$
\begin{aligned}
&(\ell_4, m_4) \\
&(\ell_2, m_2) \longrightarrow \bigcirc \longrightarrow (\ell_3, m_3) \qquad = \qquad \prod_{i=1}^{4} \left( \sqrt{\frac{\pi}{2\beta}} \, I_{\ell_i + \frac{1}{2}}(\beta) \right)^{1/2} \\
&\hspace{3.2cm}(\ell_1, m_1) \qquad\qquad\qquad \times \int \mathrm{d}\Omega_k \, Y_{\ell_1, m_1}(\theta_k, \phi_k) \, Y_{\ell_2, m_2}(\theta_k, \phi_k) \, Y^*_{\ell_3, m_3}(\theta_k, \phi_k) \, Y^*_{\ell_4, m_4}(\theta_k, \phi_k)
\end{aligned}
\tag{19}
$$

around the vertices $k \in V$. This time though, the integral over the two continuous parameters

$$
F^{(\ell_3, m_3),(\ell_4, m_4)}_{(\ell_1, m_1),(\ell_2, m_2)} = \int \mathrm{d}\Omega_k \, Y_{\ell_1, m_1}(\theta_k, \phi_k) \, Y_{\ell_2, m_2}(\theta_k, \phi_k) \, Y^*_{\ell_3, m_3}(\theta_k, \phi_k) \, Y^*_{\ell_4, m_4}(\theta_k, \phi_k)
\tag{20}
$$

involves four spherical harmonics corresponding to the four links attached. Again, it is non-vanishing if and only if the total angular momentum is zero: This could be ensured by fusing the two incoming angular momenta ($\ell_1$ and $\ell_2$) to an intermediate value $\ell$, which is then split again into the two outgoing ones ($\ell_3$ and $\ell_4$). The coupling rules of spherical harmonics are in close connection to the spin coupling rules of $SU(2)$, which makes an $SU(2)$-symmetric tensor network ansatz again very convenient. The

precise relation reads

$$F_{(\ell_1,m_1),(\ell_2,m_2)}^{(\ell_3,m_3),(\ell_4,m_4)} = \sum_\ell f_{\text{CG}}(\ell_1,\ell_2,\ell)\, f_{\text{CG}}(\ell_3,\ell_4,\ell) \ \times \qquad\qquad , \qquad (21)$$

where the numerical factors converting between the two representations are given by

$$f_{\text{CG}}(\ell_1,\ell_2,\ell) = \sqrt{\frac{(2\ell_1+1)(2\ell_2+1)}{4\pi(2\ell+1)}}\, \langle \ell_1\,0\,\ell_2\,0 \,|\, \ell\,0 \rangle\,, \qquad (22)$$

and the fusion tree represents the Clebsch-Gordan coefficients. Besides the usual constraint $m_1 + m_2 = m_3 + m_4$, we have here an additional one dictated by the reflection symmetry of the $O(3)$ group, namely $\ell \mod 2 = (\ell_1 + \ell_2) \mod 2 = (\ell_3 + \ell_4) \mod 2$. Ultimately, the partition function for the Heisenberg model on the two-dimensional square lattice can be constructed by the contraction of local four-index tensors of the form

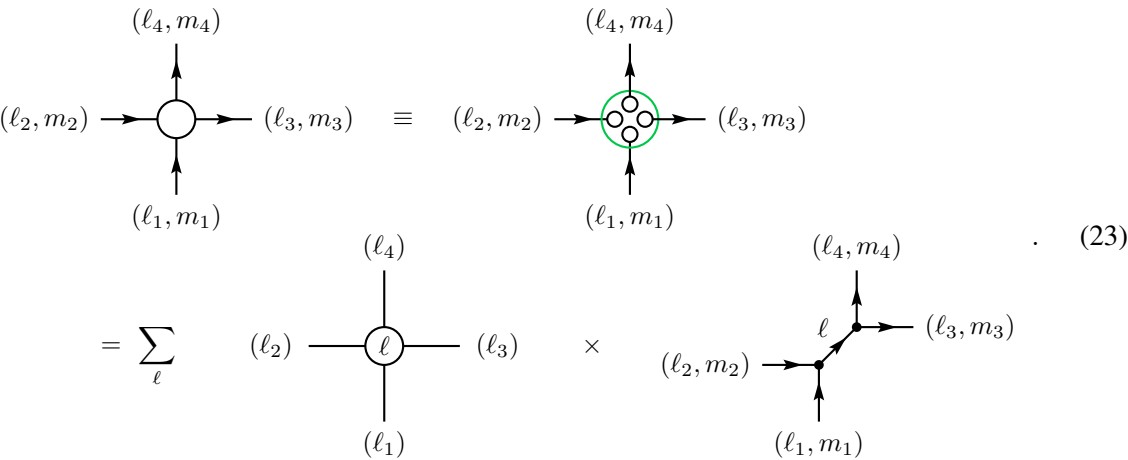

$$(23)$$

The degeneracy tensors encompass this time not only the numerical Boltzmann factors as in the one-dimensional case, but also the conversion factors of Eq. (22) and are hence given by

$$\qquad = \ \prod_{i=1}^{4}\left(\sqrt{\frac{\pi}{2\beta}}I_{\ell_i+\frac{1}{2}}(\beta)\right)^{1/2} \times f_{\text{CG}}(\ell_1,\ell_2,\ell)\, f_{\text{CG}}(\ell_3,\ell_4,\ell)\,. \qquad (24)$$

The handling of $SU(2)$-symmetric tensors is implemented in an automated library, that deals with the symmetry constraints efficiently in terms of fusion trees [63]. Some further remarks are in order. The representation of the partition function as a tensor network will necessarily be an approximation. This

is due to the fact that the plane wave expansion of the Boltzmann terms in Eq. (8) is only exact in the limit $\ell \to \infty$. In practice, the links in our tensor network ansatz will always deal with a finite number of angular momenta $0, 1, \ldots, j_{\max}$ (we will use $\ell$ for angular momenta and $j$ for $SU(2)$ quantum numbers in our TN). This in turn means that the number of representations kept will determine the accuracy of our simulation and provides a systematic refinement parameter, as we will demonstrate in the next sections.

## 3.3 Partition function with corner transfer matrices

Let us now demonstrate how the tensor network representation is used in practice to compute the partition function. The central object for the Heisenberg model on the square lattice is the four-index tensor of Eq. (23). The full partition function is then the contraction of the infinite square lattice with a four-index tensor on every vertex. Although the construction is exact in principle, practical reasons of the tensor network ansatz force us to approximate the calculations of $\mathcal{Z}$. These approximations are controlled by the number of angular momenta used in the plane-wave expansion of the Boltzmann factors, see Eq. (8). The number of angular momenta kept is then directly related to the bond dimension of the bulk tensors. All the links of the tensor network are described in terms of $SU(2)$ quantum numbers denoted by $j_{t_j}$, where $j$ is the spin in the language of Clebsch-Gordan coefficients (or angular momentum $\ell$ in the language of spherical harmonics, with the above mentioned further restrictions to fusion rules) and $t_j$ its degeneracy (see also Ref. [63] for details about $SU(2)$ symmetry in TNs). Keeping only the lowest value of $j_B = 0_1$ amounts to a bond dimension of $\chi_B = 1$, i.e., a product state. Naturally, as we increase the bond dimension by keeping more spin quantum numbers,

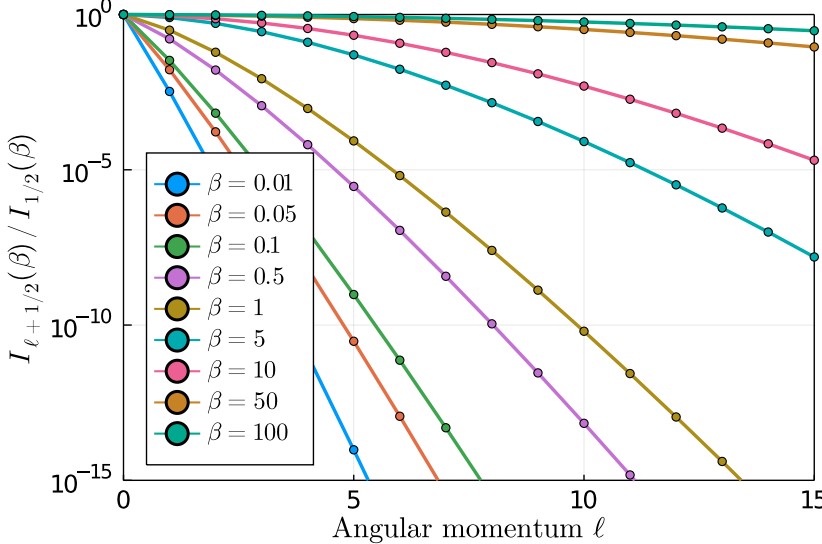

Figure 1: Decay of the Bessel functions $I_{\ell+1/2}(\beta)$ normalized to $I_{1/2}(\beta)$ as a function of the angular momentum $\ell$, for several values of the inverse temperature $\beta = 1/T$.

the accuracy of the approximation of $\mathcal{Z}$ increases as well. Here, $\chi_B$ counts the number of terms in the plane-wave expansion of the Boltzmann factors, such that a tensor network with bond dimension $\chi_B$ has spins $j_B = [0_1, 1_1, \ldots, (\chi_B - 1)_1]$ on the links. Notice also that $\chi_B$ is the symmetric bond dimension, while the effective bond dimension is larger due to the intrinsic degeneracy of $SU(2)$ quantum numbers (a spin $j$ is $(2j+1)$-fold degenerate, see also Table 1). In our study we consider the

temperature range of $T \in [0.01, 100]$. For each individual temperature we initialize an infinite tensor network with a single-site unit cell tensor according to Eq. (23) for bond dimensions up to $\chi_B = 6$, i.e., with a maximal spin of $j_B^{\max} = 5$ on the links. It is expected, that this bond dimension is more than sufficient for large temperatures. However, at low temperatures the approximations due to a finite $\chi_B$ will become strong, as forecasted by the behaviour of the Bessel functions in Fig. 1. While the Bessel functions drop off very quickly for large $T$ and $\chi_B = 4$ already yields a relative accuracy of $\approx 10^{-5}$, the bond dimension would have to be $\chi_B \sim 50$ to achieve the same level of accuracy at low $T$. The rather small chosen value of $\chi_B = 6$ is due to limited computational power, especially when actually contracting the infinite TN, as we will elaborate next. The contraction of the infinite two-dimensional square lattice can only be computed approximately, for instance using a standard *corner transfer matrix* (CTM) procedure. This is basically a power method in which the tensor for the partition function is absorbed into so-called *environment tensors*, until they converge to an environment fixed-point. The contraction of the infinite lattice is then approximated by eight fixed-point tensors $\{C_1, C_2, C_3, C_4\}$ and $\{T_1, T_2, T_3, T_4\}$, according to

$$
\mathcal{Z} = \begin{pmatrix} \vdots & \vdots & \vdots \\ \cdots \bigcirc - \bigcirc - \bigcirc \cdots \\ \cdots \bigcirc - \bigcirc - \bigcirc \cdots \\ \cdots \bigcirc - \bigcirc - \bigcirc \cdots \\ \vdots & \vdots & \vdots \end{pmatrix} \approx \begin{matrix} C_1 & T_1 & C_2 \\ T_4 & & T_2 \\ C_4 & T_3 & C_3 \end{matrix} \tag{25}
$$

for a translationally invariant system with a single-site unit cell. Arrows are omitted here for better clarity. Since the indices of the bulk tensors (the white tensors in Eq. (25)) carry only integer spin representations, the CTM environment bond indices can generally carry integer, half-integer or mixed, i.e., integer and half-integer spin representations. The natural choice of only integer spin irreducible representations worked out to be the best one, with smooth convergence of the environment tensors. We notice here that this could be interpreted as the full $O(3)$ symmetry being preserved, without room for spontaneous breaking of the $\mathbb{Z}_2$ subgroup: The latter would instead result in half-integer, $SU(2)$-like irreducible representations in the environment tensors. The CTM procedure necessarily introduces a second approximation in the calculation of the partition function. It stems from the contraction of the two-dimension tensor network, which can not be performed exactly without an exponential growth of the simulation time. The approximations are controlled by the environment bond dimension $\chi_E^{\mathrm{eff}}$, that is the bond dimension along the perimeter of the CTM tensors in Eq. (25). For each fixed $\chi_B$, the environment bond dimension should be increased up to the point, where the environment tensors are converged in $\chi_E^{\mathrm{eff}}$. In this case the approximations of the contraction of the infinite two-dimensional lattice are insignificant and the only approximation is the one of the bulk tensors, $\chi_B$. However, convergence in $\chi_E^{\mathrm{eff}}$ is only achieved for rather small $\chi_B$, while approximations in $\chi_E^{\mathrm{eff}}$ remain for larger $\chi_B$ due to limited computational power. In our simulations we work with different environment bond dimensions between $\chi_E^{\mathrm{eff}} = 100$ and $\chi_E^{\mathrm{eff}} = 1500$. In contrast to the bulk bond dimension $\chi_B$, these are now effective bond dimensions. The spin sectors on the CTM environment tensors are fixed to be integer but the truncation procedure automatically adapts to the most relevant sectors. As an example, we lists the actual quantum numbers for the $SU(2)$-symmetric CTM tensors at $T = 0.01$ in Table 1, where the (effective) bond dimension of the environment tensors $\chi_E^{\mathrm{eff}}$ is broken down into its symmetric bond dimension $\chi_E$ and quantum numbers[2]. As expected, for larger $\chi_B$ and hence higher $j_B^{\max}$, the algorithm also keeps more spin sectors on the CTM bond indices.

---

[2]Notice that $\chi_E^{\mathrm{eff}}$ refers to the theoretical maximal bond dimension. Due to the preservation of $SU(2)$ multiplets in the tensor network truncation procedures, the actual bond dimension can be slightly less in practice.

| bulk tensors | | | env. tensors | | |
|:---:|:---:|:---:|:---:|:---:|:---:|
| $\chi_B$ | $j_B^{\max}$ | $\chi_B^{\mathrm{eff}}$ | $\chi_E$ | $j_E^{\max}$ | $\chi_E^{\mathrm{eff}}$ |
| 2 | 1 | 4 | 304 | 6 | 1500 |
| 3 | 2 | 9 | 204 | 9 | 1500 |
| 4 | 3 | 16 | 159 | 12 | 1500 |
| 5 | 4 | 25 | 135 | 14 | 1500 |
| 6 | 5 | 36 | 116 | 16 | 1500 |

Table 1: Symmetric bond dimension ($\chi$), maximal spin $j^{\max}$ and effective bond dimensions ($\chi^{\mathrm{eff}}$) for the tensor indices of the bulk tensors of the partition function, and for the CTM environment indices at temperature $T = 0.01$.

### 3.4 Thermodynamic quantities from the partition function

The partition function can be used to compute all the key properties of a thermodynamic system. In particular, it gives access to the free energy $F$, the internal energy $U$ and the thermodynamic entropy $S$. In this section we will compute those quantities directly from the tensor network representations described in Section 3.1 and Section 3.2. Notice that we employ a notation where the quantities are always defined per lattice site, so that we avoid obvious divergences in the thermodynamic limit.

#### 3.4.1 One-dimensional linear chain

In one spatial dimension, the partition function for the Heisenberg model has been shown to be

$$
\mathcal{Z} \quad = \quad \cdots \ \rightarrow\!\!\bigcirc\!\!\rightarrow\!\!\bigcirc\!\!\rightarrow\!\!\bigcirc\!\!\rightarrow\!\!\bigcirc\!\!\rightarrow \ \cdots \quad , \tag{26}
$$

where the $SU(2)$-symmetric matrices on each lattice site are constructed according to Eq. (14). Using the tensor network construction we can directly compute the free energy per site

$$
F(\beta) = -\lim_{N\to\infty} \frac{1}{N} \frac{\ln \mathcal{Z}_N(\beta)}{\beta} = -\frac{\ln \lambda(\beta)}{\beta}, \tag{27}
$$

where $\lambda(\beta) = \sinh(\beta)/\beta$ is the ($\ell = 0$) leading eigenvalue of the transfer matrix. This matches the exact solution for the free energy, since there are no approximations present in the one-dimensional construction (see Section 3.1).

One quantity we can directly compute from $F(\beta)$ is the thermodynamic entropy per site, given by $S(\beta) = -\partial F/\partial T$. The free energy $F$ and the thermodynamic entropy $S$ are shown in Fig. 2. It is known that thermodynamic potentials are concave functions of their intensive variables and convex functions of their extensive variables [73]. This is indeed found for the free energy $F$, for which the logarithmic temperature scale skews the picture. However, due to the permanent positive slope, the thermodynamic entropy becomes negative. This is somewhat surprising and quite counter-intuitive, and the entropy even diverges as $T$ approaches zero like $S(T \to 0) \to -\infty$. This behaviour is however typical for classical spin models [71,73] with continuous degrees of freedom[3], and *not* a misfeature of the tensor network construction. Now that we have validated our tensor network construction against an exactly solvable model, we will move our focus to the more interesting two-dimensional square lattice scenario.

---

[3]The same behaviour is found for the $O(2)$-case (i.e., XY), with the leading eigenvalue $\lambda(\beta) = I_0(\beta)$ [33, 34].

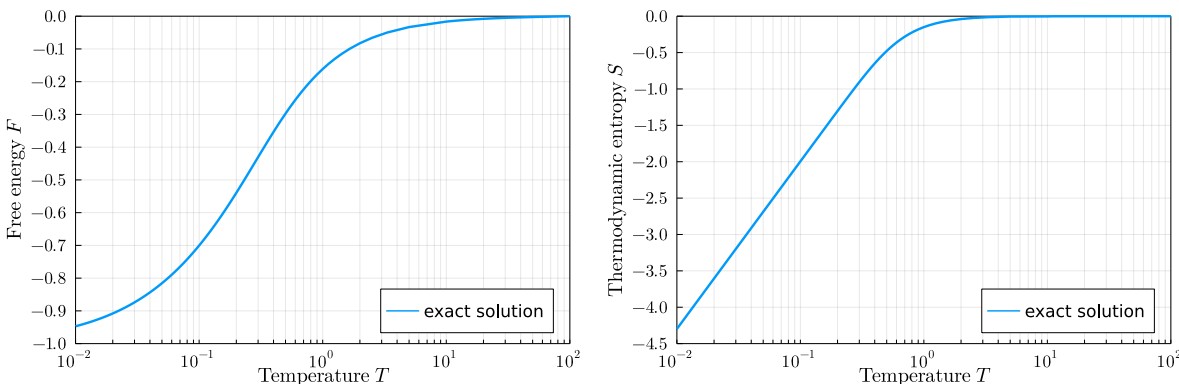

Figure 2: Free energy $F$ and thermodynamic entropy $S$ per site for the one-dimensional classical Heisenberg chain. The negative thermodynamic entropy is a particular feature of the model.

### 3.4.2 Two-dimensional square lattice

Similarly to the one-dimensional case, we define the partition function per site as

$$\ln(z(\beta)) = \lim_{N_x N_y \to \infty} \frac{1}{N_x N_y} \ln(\mathcal{Z}(\beta)) . \tag{28}$$

We can compute it again from the dominant eigenvalue of a transfer operator (an infinite row or column of four-index tensors), and then express the latter itself in terms of the dominant eigenvalue of its constituent transfer matrix in the other direction (column or row, respectively). Without going into the details, $z(\beta)$ can thus be expressed in terms of the CTM tensors in such a way that their normalization fully cancels, see for instance Ref. [74]:

$$z(\beta) = \left( \begin{array}{c} \end{array} \right) \bigg/ \left( \begin{array}{c} \end{array} \right) . \tag{29}$$

We can now proceed to extract the free energy $F$ and the entropy $S$ per site. In both panels of Fig. 3 we show the data for all available bulk bond dimensions at the maximum environment bond dimension $\chi_E^{\text{eff}} = 1500$. The overall behaviour of $F$ and $S$ resembles the one of the one-dimensional Heisenberg chain, and the negative entropy is again not a misfeature of our tensor network method. However, in the two-dimensional setting we are dealing with approximations in $z(\beta)$ and hence in the thermodynamic quantities. Based on the truncation of the bulk bond dimension $\chi_B$ and the behaviour of the Bessel functions for large $\beta$ (refer to Section 3.3), we identify low temperatures, i.e., $T \leq 0.2$, as the regime in which our approximations become strong, and we highlight this with a grey background. In this regime a much larger bond dimension $\chi_B$ would be desirable for all links in the TN. It might well be that the approximations lead to larger error bars, while the data points could still exhibit a good trend towards their actual values. However, the concrete impact of the $\chi_B$- truncation is difficult to assess. The choice of the boundary value of $T = 0.2$ is supported by the spread of data curves in Fig. 3, and even clearer confirmed by further analysis in subsequent sections. A lighter gray highlighted area for $0.2 < T \leq 0.5$ is introduced to indicate the region where our tensor network data seem to be not

dramatically spread with respect to $\chi_B$, but an extrapolation in the environment bond dimension $\chi_E$ looks crucial to fully capture the physical behaviour (see Section 4).

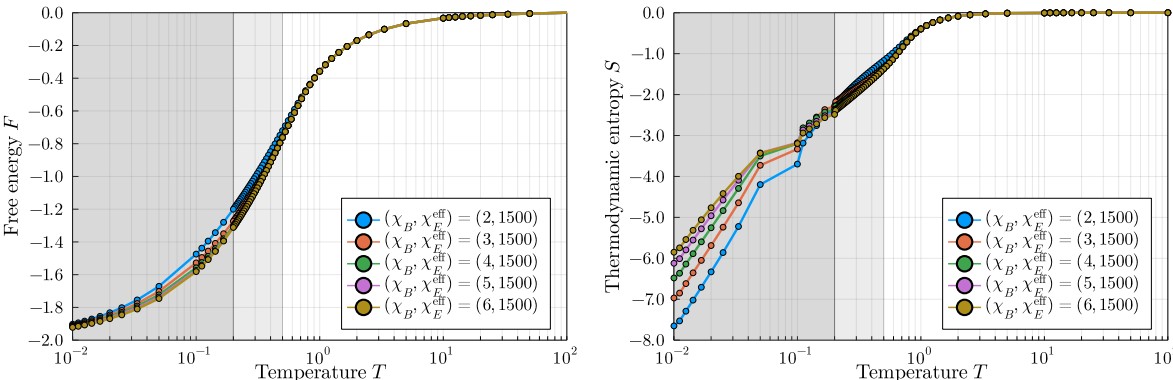

Figure 3: Free energy $F$ and thermodynamic entropy $S$ per site for the two-dimensional classical Heisenberg model on a square lattice. Both quantities are directly computed from the tensor network construction approximating the partition function according to Eq. (29).

## 3.5 Entanglement analysis

One of the many advantages of using tensor network-based approaches is that we can comparably easily get access to entanglement properties almost directly by construction. These quantities can then be used to describe and detect phases as well as the phase transitions between them. For instance, in one spatial dimension, the entanglement spectrum, i.e., (minus log of) the eigenvalue spectrum of a reduced density matrix, can be extracted from a bipartition of a *matrix product state* (MPS). This can then be analysed to spot signatures of distinct topological and trivial phases of matter protected by symmetries [75–80]. It can also be used to compute the entanglement entropy, as well as Rényi entropies, whose scaling behaviour can then be used to locate phase transitions. Similarly, in two dimensions, the environment tensors obtained from the CTM procedure after contracting the infinite two-dimensional lattice can also be used to compute different quantities that are associated to entanglement properties of some underlying quantum model. For the case at hand in this paper, this allows us to determine some characteristics of the model in different temperature regimes. To be more specific, the model that we study here is two-dimensional classical, but the associated "entanglement" computed from the CTM tensors is that of the associated dual one-dimensional quantum model in the same universality class [81]. Therefore, even though the model we study here is purely classical, these quantum-inspired quantities, nonetheless, give us valuable information also about the classical correlations present in our system, which can then be used to characterize phases and their transitions. We describe them in more detail below.

### 3.5.1 Corner entropy

The environment fixed-point tensors can be directly used to compute the corner entropy $S_C$, a quantity that is known to contain information about the universal properties of the model at hand [81]. As such it can help to pinpoint phase transitions. The corner entropy is related to the entanglement entropy of a bipartition for some infinite one-dimensional quantum Heisenberg model, due to the general mapping between $D$-dimensional classical statistical mechanics models and $(d+1)$-dimensional quantum

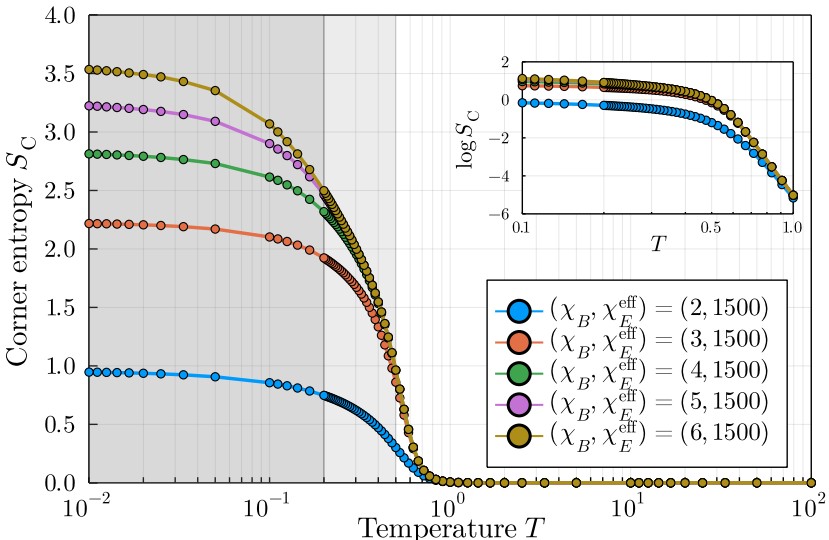

Figure 4: Corner entropy $S_C$ of the fixed-point CTM corner tensors for different maximal values of the spin in the plane wave expansion and the virtual bond indices. The inset shows the region around $T \approx 0.5$.

models [21]. It can be computed from an eigenvalue decomposition of the matrix $C := C_1 \cdot C_2 \cdot C_3 \cdot C_4$, whose graphical notation in the tensor network language is

$$C \quad = \quad \begin{array}{cc} C_1 & C_2 \\ & \\ C_4 & C_3 \end{array} \quad . \tag{30}$$

The eigenvalues $\lambda_\alpha$ appear with different degeneracies due to the block-diagonal structure of the $SU(2)$-symmetric matrix $C$. The corner entropy can then be computed using

$$S_C = -\sum_\alpha \lambda_\alpha \log \lambda_\alpha \,, \tag{31}$$

where the degeneracies of the eigenvalues $\lambda_\alpha$ in each spin sector need to be taken into account. Results for all available bond dimensions up to $\chi_B = 6$ are shown in Fig. 4. For temperatures $T \geq 0.2$ the corner entropy is essentially converged in the virtual spin irreducible representations, however, in the low-temperature regime for $T < 0.2$ more quantum numbers would be needed to achieve convergence, as indicated by the separation of the data points and expected from the properties of the Bessel functions (cf. Fig. 1). The behaviour of the corner entropy is a first indication of different physical properties in the low- and high-temperature regime. While it hints at an uncorrelated phase at high temperature, the increasing correlations towards low temperature already indicate the emergence of a highly-correlated low-temperature phase. To be precise, in the figure we can distinguish three different regimes: above $T \sim 0.6$ (vanishing entropy), between $T \sim 0.6$ and $T \sim 0.2$ (linear behaviour in $T$ and converged in $\chi_B$), and below $T \sim 0.2$ (weak dependence on $T$ and $S_C$ increasing with $\chi_B$).

### 3.5.2 Geometric entanglement

As we have mentioned in the introduction (Section 1), some past works have hinted at the possible presence of a BKT-like transition at low temperatures – incidentally compatible with the change in the slope of the corner entropy in Fig. 4. Such kind of transitions are however challenging to detect numerically: Most of the usual figures of merit fail to capture them in a clear-cut way [82]. In order to confirm or disqualify such a transition, we resort here to the *geometric entanglement (GE)* for the boundary MPS obtained from the CTM procedure. It is indeed known that the global geometric entanglement is able to capture elusive phase transitions [82] even in the case where no indications are found in other entanglement measures such as the entanglement entropy [83]. The geometric entanglement describes the proximity of an entangled quantum state vector $|\psi\rangle$ to the closest product state vector $|\phi\rangle$ via the overlap

$$\Lambda_{\max} := \max_{\phi} |\langle\phi|\psi\rangle| \ . \tag{32}$$

The geometric entanglement per site is then defined as

$$\epsilon = \lim_{N\to\infty} -\frac{\log \Lambda^2}{N} \ . \tag{33}$$

We computed the geometric entanglement per site for the boundary MPS obtained after contracting a half-infinite two-dimensional lattice of the partition function tensors. As such, this MPS corresponds to the ground state of the associated dual one-dimensional quantum model (with entanglement entropies such as the corner entropy computed in the previous section). This MPS can be easily built from our CTM technique, just by considering an infinite MPS with tensor $T_1$ at every site. In Fig. 5 we

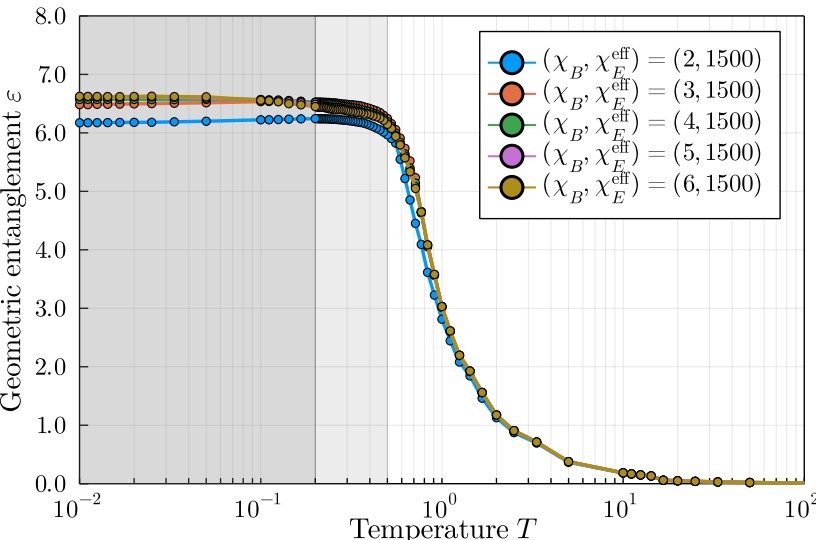

Figure 5: Geometric entanglement per site $\epsilon$ for the boundary MPS obtained from the CTM procedure. We observe a smooth crossover, which is incompatible with a phase transition, even of the BKT type, as explained in Ref. [82].

show the geometric entanglement per site computed for such boundary MPSs. For large temperatures the proximity to a product state is almost perfect, which yields a vanishing GE. For lower temperatures

it becomes increasingly difficult to approximate the boundary MPS with a product state, as already expected from the behaviour of the corner entropy. The GE shows however a crossover but not any sign of a phase transition. As shown in Ref. [82], the geometric entanglement per site shows a clear peak when crossing a BKT transition, being in fact very sharp and similar to a level crossing. This is due to the fact that it is a multipartite measure of entanglement, and therefore can capture the subtleties of global changes in the structure of the physical system, as opposed to other entanglement and correlation measures that are essentially more local. In our case, as we have discussed above, we see no peak whatsoever, and just a smooth crossover.

## 4   Computation of expectation values

In this section, we will show how to compute expectation values of observables, both local and at a distance, and discuss how to extract the model's correlation length, both from fitting correlations as well as directly from the tensor network formalism. The aim is to explore the full temperature regime and try to shed light onto one of the most important core issues of the model, namely that of a finite-temperature phase transition. In particular, we will apply a recently introduced scaling hypothesis for finite bond dimension data in order to better estimate the true correlation length: This will help avoiding to over-interpret the bare data at low temperatures, where numerical approximations intrinsic to the method might cap the (very large) correlation length. Nonetheless, at first sight, the data seem to fall well onto the typical fitting form of BKT-like transitions at the before mentioned finite temperature $T_c \simeq 0.5$, although no other signatures thereof are found (see also Section 3.5.2). Conversely, we show that our data are overlapping with reference ones in the lattice gauge theory community (in the region where both are available): By performing the same kind of scaling to infinite correlation length customary in that context, we assess the consistency of our data with the asymptotic freedom hypothesis, i.e., with no finite-$T$ transition. The puzzle thus still remains open, but we foresee that massive tensor network simulations in the regime $T \in [0.1, 0.5]$ would considerably help to lift the mystery in the near future. We plan to take up this formidable challenge in upcoming works.

### 4.1   Construction of observables

The computation of expectation values is relatively straightforward in our present formulation of the classical partition function. An $n$-site observable results in the contraction of a tensor network with $n$ local tensors modified in a non-trivial way, dictated by the decomposition of the observable in terms of the group characters, plus possibly a few more trivial tensors mediating the correlation without changing their own character. Let us see in practice what this means for certain relevant cases up to $n = 3$. In general, an $n$-site observable is given by

$$\left\langle O(\hat{S}_1, \ldots, \hat{S}_n) \right\rangle = \frac{1}{\mathcal{Z}} \int \mathcal{D}\Omega \, \mathrm{e}^{-\beta E(\{\hat{S}\})} \, O(\hat{S}_1, \ldots, \hat{S}_n) \,, \tag{34}$$

where $\hat{S}_1, \ldots, \hat{S}_n$ act on $n$ different lattice sites and not necessarily on site $1, \ldots, n$. Naturally, the infinite contraction of the tensor network surrounding the $n$-sites with non-trivial modifications can again be approximated by the fixed-point environment tensors.

   The only non-trivial one-site observable is in principle $O(\hat{S}_k) = \hat{S}_k$, i.e., the magnetization per site. However, since this operator is not $SU(2)$- (resp. $O(3)$-) invariant, the local magnetization has to vanish. This is not only expected for a manifestly invariant Ansatz like ours, but it is also anyway

forbidden by the Mermin-Wagner theorem, which excludes spontaneous breaking of the continuous symmetry of the Heisenberg model.

As a non-trivial two-site observable we will consider the spin scalar product $O(\hat{S}_i, \hat{S}_j) = \hat{S}_i \cdot \hat{S}_j$. When the operator acts on nearest-neighbours, it measures the (negative) bond energy, otherwise it measures generic spin-spin correlations at a distance $|i - j|$. The spin scalar product can be written in terms of spherical harmonics acting on sites $i$ and $j$ as

$$\hat{S}_i \cdot \hat{S}_j = \frac{4\pi}{3} \sum_{m=-1,0,+1} Y^*_{1,m}(\theta_i, \phi_i) \, Y_{1,m}(\theta_j, \phi_j) \, . \tag{35}$$

This additional spherical harmonic introduces a fifth tensor index carrying a spin-1, which then calls for straightforward modifications in the underlying fusion trees and, consequently, in the conversion factors to Clebsch-Gordan coefficients. Assuming, without loss of generality, to measure the spin-spin correlations along horizontal bonds, the whole expression reads

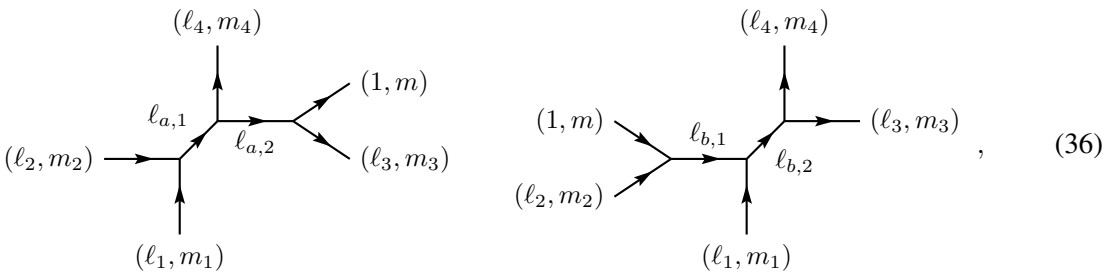

$$\tag{36}$$

$$
\begin{array}{c}
(\ell_4) \\
(\ell_2) -\!\!\!\square\!\!\!\begin{array}{c}(1)\\(\ell_3)\end{array} \\
(\ell_1)
\end{array}
\quad = \quad
\sqrt{\frac{4\pi}{3}} \prod_{i=1}^{4} \left( \sqrt{\frac{\pi}{2\beta}} I_{\ell_i + \frac{1}{2}}(\beta) \right)^{1/2}
\quad , \tag{37}
$$
$$\times f(\ell_1, \ell_2, \ell_{a,1}) \, f(\ell_{a,2}, \ell_4, \ell_{a,1}) \, f(\ell_3, 1, \ell_{a,2})$$

$$
\begin{array}{c}
(\ell_4) \\
\begin{array}{c}(1)\\(\ell_2)\end{array}\!\!\!\square\!\!\!- (\ell_3) \\
(\ell_1)
\end{array}
\quad = \quad
\sqrt{\frac{4\pi}{3}} \prod_{i=1}^{4} \left( \sqrt{\frac{\pi}{2\beta}} I_{\ell_i + \frac{1}{2}}(\beta) \right)^{1/2}
\quad . \tag{38}
$$
$$\times f(\ell_2, 1, \ell_{b,1}) \, f(\ell_1, \ell_{b,1}, \ell_{b,2}) \, f(\ell_3, \ell_4, \ell_{b,2})$$

The two operators acting on vertical bonds are defined in complete analogy, and such constructions

can be straightforwardly extended to three-sites observables, like for instance

$$\left\langle O(\hat{S}_i, \hat{S}_j, \hat{S}_k) \right\rangle \quad = \quad$$ 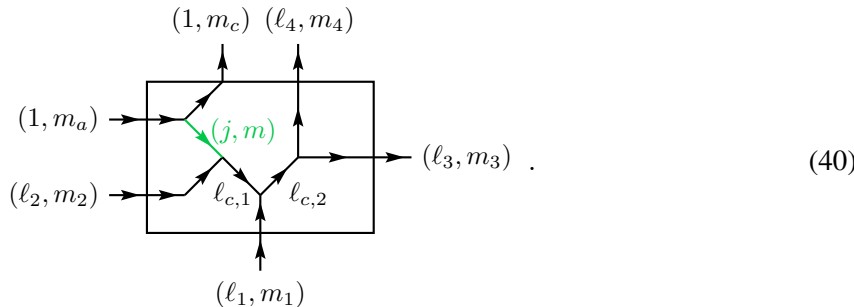 $$\tag{39}$$

The central tensor (the bottom right one in this example) has two additional spin-1 channels, which connect to the scalar product operators on both sites respectively. The underlying fusion tree determines which expectation value gets measured,

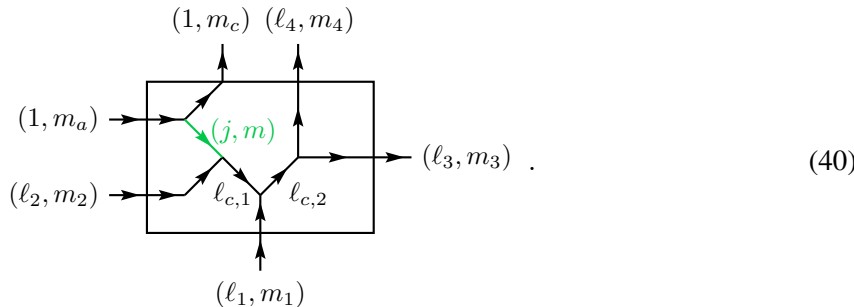

$$\tag{40}$$

Setting the eminent internal spin to $j = 0$ yields an identity on the central site, and therefore next-to-nearest neighbour spin correlations $\langle \hat{S}_i \cdot \mathbb{I}_j \cdot \hat{S}_k \rangle$; replicating this an arbitrary number of times can therefore be used to generate spin scalar product at larger distances. Choosing $j = 1$ instead generates the interesting triple product $O(\hat{S}_i, \hat{S}_j, \hat{S}_k) = \langle \hat{S}_i \cdot (\hat{S}_j \times \hat{S}_k) \rangle$. Measuring this observable on all triangles on a square lattice plaquette could be useful in uncovering twisted spin configurations, similar to those that appear in magnetic Skyrmion systems. Noticeably, this quantity is $SU(2)$-invariant but breaks the additional discrete $\mathbb{Z}_2$ reflection symmetry of the $O(3)$ group. Therefore, it could in principle exhibit finite expectation values without violating the Mermin-Wagner theorem.

## 4.2   Bond energy

We now focus on the bond energy, which is computed using the scalar product operators introduced previously in Eqs. (36)-(38) and setting $|i - j| = 1$ as

$$-\left\langle \hat{S}_i \cdot \hat{S}_j \right\rangle \quad = \quad -\frac{1}{\mathcal{Z}} \left( \quad \right) \quad \approx \quad - \quad \Big/ \quad . \tag{41}$$

Here the square tensors are the ones generating the scalar product, while all the other ones are the analytic tensor for the partition function as in Eq. (23). Notice the additional negative sign that appears

due to the ferromagnetic coupling in the Hamiltonian. Using the fixed-point environment tensors computed with the CTM procedure, the evaluation of the expectation value simplifies to the evaluation of the two rightmost tensor network diagrams. The results are shown in Fig. 6 for the accessible bond dimensions. Due to the spatial isotropy of the Hamiltonian and the tensor network construction, the energy along horizontal and vertical links is identical. The behaviour of $\langle \hat{S}_i \cdot \hat{S}_j \rangle$ for a fixed $\chi_B$ hardly

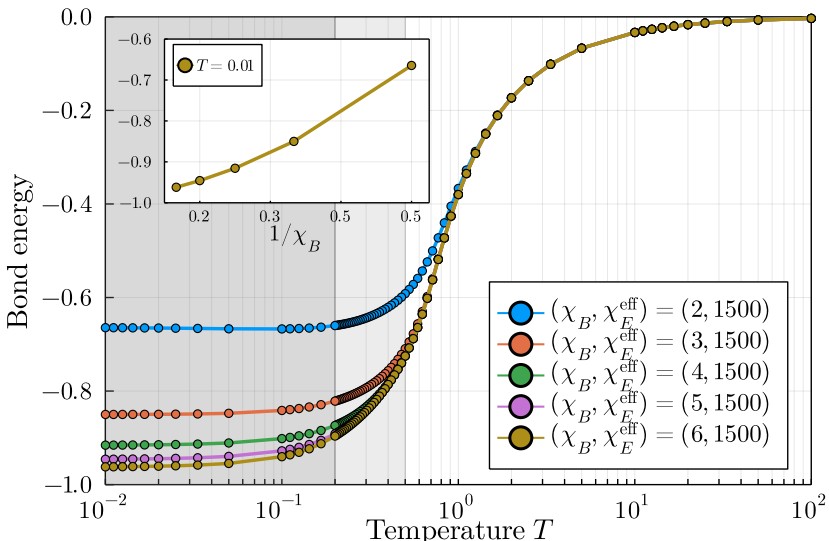

Figure 6: Bond energy $-\langle \hat{S}_i \cdot \hat{S}_j \rangle$ on horizontal and vertical nearest-neighbour links. The inset shows the convergence of the bond energy vs. the inverse bulk bond dimension at $T = 0.01$. As $T \to 0$, the relative weights of all angular momenta in Eq. (24) tend to become equal, so that the only sensible scaling indeed seems to be against the (inverse) number of the allowed ones.

depends on the environment bond dimension, so that the results are converged in $\chi_E^{\mathrm{eff}}$. This is due to the very local support of the operators and not the case for general operators or global properties.

As expected for large temperatures, the spins are fully disordered and the bond energy approaches 0 for $T \to \infty$. In contrast, for low temperatures a tendency towards ferromagnetic alignment develops and results in a bond energy tending to $-1$ for $T \to 0$. The data curves for different $\chi_B$ nicely overlap down to temperatures of $T \approx 0.2$. As we have argued before, at this point our approximations due to limited bond dimension become sizeable. However, the expected behaviour is recovered by increasingly larger values of the virtual spin $j_{\mathrm{max}}$, i.e., system bond dimension $\chi_B$.

## 4.3 Spin-spin correlations: The tensor network data and a finite-$T_c$ hypothesis

The bond energy in Section 4.2 can be seen as being a specific case of general spin-spin correlations $\langle \hat{S}_i \cdot \hat{S}_j \rangle$ at a distance $|i - j|$ for lattice sites $i$ and $j$

$$\left\langle \hat{S}_i \cdot \hat{S}_j \right\rangle = \qquad \Bigg/ \qquad . \tag{42}$$

Here the leftmost and rightmost tensors are the ones for the scalar product from Eqs. (36)-(38), while the ones in between act as identities and simply loop through the interaction (this is achieved by setting $j = 0$ in the fusion tree, see Eq. (40) and surrounding text). It is also important to stress that

the denominator must be computed via the full contraction over the same finite distance $|i - j|$ as the numerator, in order for all eigenvalues of the transfer matrix to be accounted in the same way. As expected for a ferromagnetic model, the correlations have matching signs. For large temperatures, the results nicely match an exponential decay with a short correlation length $\xi$, which can therefore be quite comfortably extracted by standard fitting procedures. For low temperatures, instead, it becomes increasingly difficult to distinguish the correlation behaviour from a quasi-algebraic decay. Very large separations between the spins would be needed due to a rapid increase of $\xi$, but these are computationally expensive to be performed according to Eq. (42).

The tensor network representation of the partition function provides however a direct and much cleaner access to compute the correlation length, i.e. via the eigenvalue spectrum $\lambda_\alpha$ of the transfer matrix $\mathcal{T}$ of the boundary MPS. The transfer matrix is constructed out of the infinite half-column CTM tensors $T_1$ and $T_3$ (or, equivalently, from $T_2$ and $T_4$ in the other direction) according to

$$\tag{43}$$

The correlation length is then computed as [30, 31]

$$\frac{1}{\xi} = -\log \left| \frac{\lambda_2}{\lambda_1} \right| = -\log |\lambda_2| \, , \tag{44}$$

where $\lambda_1$ and $\lambda_2$ are the two largest eigenvalues, and the rightmost expression corresponds to a normalized spectrum such that $\lambda_1 = 1$. The additional advantage of this method is that the extracted $\xi$ is automatically the largest one in the system, irrespective of which correlation function gives rise to it. In Fig. 7 we show the correlation length $\xi$ for different values of $\chi_B$ at the largest available environment bond dimension of $\chi_E^{\text{eff}} = 1500$. The inset in Fig. 7 shows a good agreement of our data with Monte Carlo simulations [84], which are among the reference ones for the lattice gauge theory community. We will see later how to reconcile them with the predictions by asymptotic freedom, as it is customary in that context. Here, instead, we first point our attention to the apparently striking consistency of the data with a fit of the kind

$$\ln \xi(T) = a + \frac{b}{\sqrt{T - T_c}} + c\sqrt{T - T_c} \, , \tag{45}$$

with $T_c = 0.509 \pm 0.003$ and $c \ll 1$, which is commonly taken as a pristine signature of a Berezinskii-Kosterlitz-Thouless transition [16, 17]. We try to heal finite bond dimension effects by adopting a scaling procedure recently introduced to cope with finite-entanglement in quantum tensor network simulations [40, 41, 85], as described in the following paragraph and displayed in Fig. 8. Thereby, the correlation length can be confidently extrapolated down to a bit below $T = 0.5$, which is shown in the main panel of Fig. 7 and the actual values are reported in Table 2. This indicates, most probably, that the BKT-like fit of Eq. (45) is not really reliable, though intriguing at first sight away from the putative $T_c$. It is however not excluded, from the numerical data at disposal, that a divergence sets in at lower temperatures.

Our tensor network includes two approximation parameters, $\chi_B$ and $\chi_E^{\text{eff}}$. However, for every fixed $\chi_B$ the second approximation in the CTM environment tensors could in principle be eliminated for

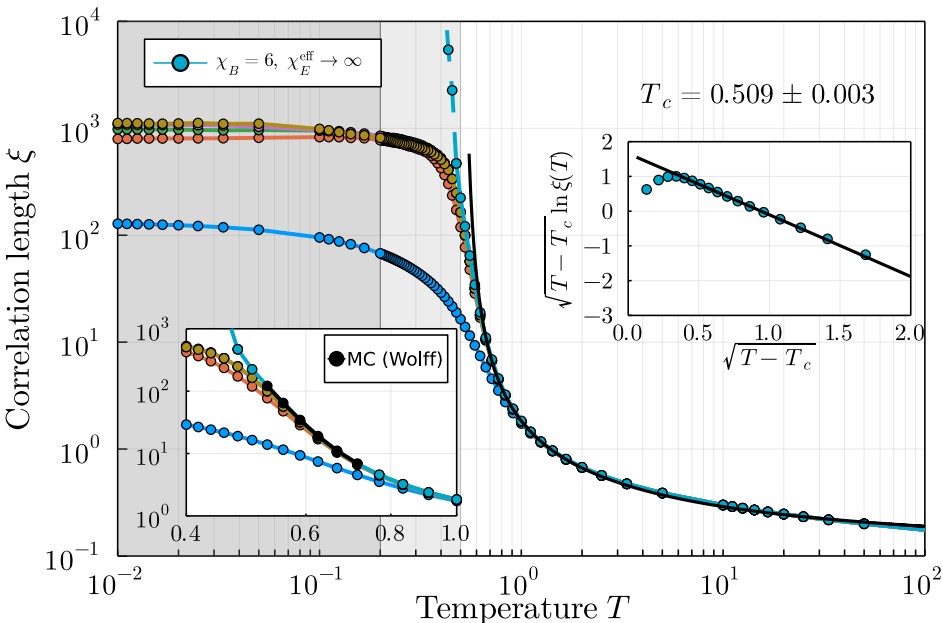

Figure 7: The largest correlation length in the system as extracted from Eq. (44) for $\chi_B = 2$ to $\chi_B = 6$ at the largest available environment bond dimension $\chi_E^{\text{eff}} = 1500$ (usual color legend). Additionally, we show the correlation length extrapolated to $\chi_E^{\text{eff}} \to \infty$. Importantly, there is no drift of the temperature dependence in the bond dimension $\chi_B$. The excellent agreement with the exponential fit of Eq. (45), down to $T = 0.6$ at least, seems at first sight to indicate a BKT-like transition at a finite temperature $T_c \approx 0.51$. The choice of the exponential fit is supported by the top-right inset, which shows the correlation length rescaled to $\sqrt{T - T_c} \ln \xi(t)$ vs. $\sqrt{T - T_c}$. However, even the extrapolated correlation length starts to deviate from a BKT-like divergence for $T \sim T_c$. The bottom-left inset shows excellent overlap with the Monte Carlo data presented in Ref. [84], which are usually employed to claim agreement with the asymptotic freedom description of the underlying field theory. For a thorough discussion, see the main text.

$\chi_E^{\text{eff}} \to \infty$, which provides us with a reliable scaling. As has been demonstrated in Refs. [41, 85], the eigenvalue spectrum of the transfer matrix $\mathcal{T}$ can be used to define a scaling parameter $\delta = \delta(\chi_E^{\text{eff}})$. The true transfer matrix eigenvalue spectrum would indeed be continuous above a possible initial gap, which in turn determines the correlation length of the system, as described above in Eq. (44). However, the necessarily finite bond dimension $\chi_E^{\text{eff}}$ naturally introduces a cutoff and therefore a maximal length scale. The parameter $\delta$ then describes the discreteness of the spectrum, i.e., its distance to a continuous spectrum. While the first gap is related to the correlation length, the second gap of $\mathcal{T}$ is already suitable to define a scaling using

$$\delta = \varepsilon_2 - \varepsilon_3 \quad \text{where} \quad \varepsilon_\alpha = -\log \lambda_\alpha \ . \tag{46}$$

A more sophisticated scaling parameter can be computed with a linear combination of the first $n$ eigenvalues, with coefficients that can be determined manually or even in an optimization procedure [41]. For our purposes, the basic definition in Eq. (46) works equivalently well. In order to perform an infinite-bond dimension extrapolation for the correlation length, we utilize a relation be-

| $\beta$ | $\chi_E^{\text{eff}}$ | | | | | | | | | MC |
|---|---|---|---|---|---|---|---|---|---|---|
| | 100 | 300 | 500 | 700 | 900 | 1100 | 1300 | 1500 | $\infty$ | |
| 1.0 | 1.77 | 1.79 | 1.80 | 1.80 | 1.81 | 1.81 | 1.81 | 1.81 | 1.83 | - |
| 1.1 | 2.27 | 2.32 | 2.33 | 2.34 | 2.35 | 2.35 | 2.35 | 2.35 | 2.39 | - |
| 1.2 | 3.02 | 3.11 | 3.13 | 3.14 | 3.16 | 3.16 | 3.16 | 3.17 | 3.22 | - |
| 1.3 | 4.17 | 4.37 | 4.40 | 4.43 | 4.44 | 4.46 | 4.46 | 4.48 | 4.57 | - |
| 1.4 | 6.03 | 6.45 | 6.52 | 6.59 | 6.62 | 6.66 | 6.66 | 6.68 | 6.88 | 6.90 |
| 1.5 | 9.13 | 10.1 | 10.3 | 10.4 | 10.5 | 10.6 | 10.6 | 10.6 | 11.0 | 11.09 |
| 1.6 | 14.3 | 16.6 | 17.1 | 17.5 | 17.7 | 17.8 | 17.9 | 17.9 | 19.0 | 19.07 |
| 1.7 | 21.5 | 28.0 | 29.6 | 30.4 | 30.9 | 31.3 | 31.5 | 31.8 | 34.4 | 34.57 |
| 1.8 | 30.1 | 44.8 | 50.5 | 52.9 | 54.4 | 55.5 | 56.5 | 56.9 | 64.4 | 64.78 |
| 1.9 | 38.6 | 66.7 | 80.6 | 87.1 | 92.1 | 95.7 | 97.6 | 100.1 | 121.2 | 121.2 |
| 2.0 | 46.2 | 90.8 | 114.8 | 131.9 | 142.2 | 152.7 | 158.6 | 164.1 | 224.0 | - |
| 2.1 | 47.8 | 107.9 | 146.2 | 179.6 | 202.2 | 218.3 | 234.3 | 248.6 | 469.5 | - |
| 2.2 | 52.3 | 122.5 | 176.5 | 217.9 | 251.8 | 280.9 | 308.6 | 336.2 | 2267 | - |
| 2.3 | 56.0 | 133.3 | 198.1 | 247.7 | 292.1 | 334.0 | 373.9 | 407.8 | 5428 | - |
| 2.4 | 59.1 | 140.9 | 208.9 | 264.0 | 323.4 | 375.5 | 423.7 | 463.5 | 18277 | - |
| 2.5 | 61.8 | 151.3 | 222.8 | 287.1 | 348.8 | 409.7 | 461.0 | 508.9 | 74171 | - |

Table 2: Correlation length $\xi$ of the boundary MPS for different values of $\beta$ and $\chi_E^{\text{eff}}$, at the maximal bulk bond dimension $\chi_B = 6$. The values at finite $\chi_E^{\text{eff}}$ are computed from the sub-dominant eigenvalue of the transfer matrix. The extrapolated values for $\chi_E^{\text{eff}} \to \infty$ are obtained by a finite-entanglement analysis demonstrated in Fig. 8. This procedure becomes unreliable for the values highlighted in gray, as discussed in the main text. Monte Carlo data from Ref. [84] is shown for comparison.

tween $\epsilon = 1/\xi = -\log|\lambda_2|$ and $\delta$ according to

$$\epsilon = a + b \cdot \delta , \qquad (47)$$

presented in Ref. [85]. In Fig. 8 we depict the finite-entanglement scaling for $\chi_B = 6$ at different inverse temperatures $\beta$, with $\delta$ computed for all available $\chi_E^{\text{eff}}$. For small values of $\beta$ the scaling procedure yields a reliable extrapolation of $1/\xi$, as demonstrated in the lower panel for $\beta = 1.0$ and $\beta = 2.0$. Notice that a change of slope sets in at values of $\delta < 0.02$ in the central lower panel of Fig. 8. We cannot exclude that this will happen at even lower values of $\delta$ for lower temperatures (larger $\beta$), i.e., that the estimate of $\xi$ might be strongly affected by data with even larger bond dimensions $\chi_E^{\text{eff}}$. Therefore, we highlight extrapolated values for $\beta \geq 2.2$ in Table 2. Nevertheless, the scaling in the confident temperature range demonstrates a rapidly increasing correlation length, which however does *not* automatically imply a true divergence. It might as well be that a saturation sets in for large $\chi_E^{\text{eff}}$ that are currently inaccessible and therefore point at no finite-$T$ transition, but rather to a quasi-critical region at low temperatures. The absence of a finite-temperature phase transition would be consistent with the geometric entanglement analysis performed in Section 3.5.2, that seemed not to show any sign of a BKT-like transition despite its sensitivity to it.

## 4.4 Spin-spin correlations: A consistency check with asymptotic freedom

As we have recalled in Section 2, the plausibly most recognised theoretical framework to describe the classical Heisenberg model in two spatial dimensions is the one relating it to the $O(3)$ non-linear

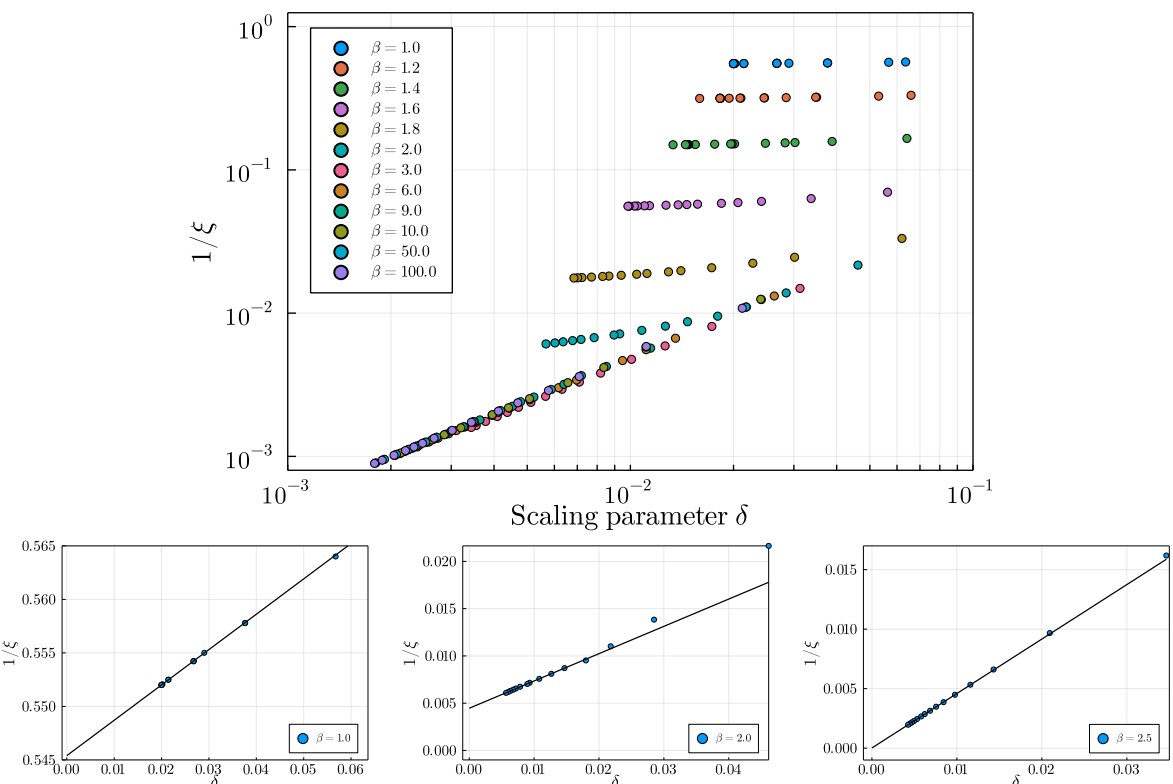

Figure 8: Top panel: Finite-entanglement scaling of $\epsilon = 1/\xi$ in $\delta(\chi_E^{\text{eff}})$ for different values of the inverse temperature $\beta$, at the maximal bulk bond dimension of $\chi_B = 6$. Bottom panel: Extrapolation of $\epsilon(\delta \to 0) = 1/\xi$ for $\beta = 1.0$ (left), $\beta = 2.0$ (middle) and $\beta = 2.5$ (right). The different vertical scales already adumbrate an unreliable extrapolation at low temperatures.

sigma model in $1 + 1$ dimensions and its asymptotic freedom. It would however be pointless to try demonstrating such mechanism from the behaviour of the correlation length itself, since no divergence of the kind $\xi(T) \simeq \exp(-1/T)$ will ever be visible in Fig. 7 within the achievable regimes. The lattice gauge theory community has instead developed a quite sophisticated consistency check with such a picture in terms of the measured spin-spin correlations [86–89], which we now illustrate and perform on our data for a maximal bond dimension of $\chi_E^{\text{max}} = 1000$.

For this sake, we should first extrapolate physical quantities to the limit of zero lattice spacing, $a \to 0$, or, equivalently, to the limit of a diverging correlation length, $\xi \to \infty$, while keeping the lattice spacing fixed as in the typical condensed matter context adopted here. This means that we should look at correlation functions at a fixed distance in terms of $\xi$, i.e.

$$C(m, \xi) := Z(\xi) \left\langle \vec{S}(x = m\xi) \cdot \vec{S}(0) \right\rangle . \tag{48}$$

We omit the vector for the position $x$ here, since we will measure correlations along horizontal bonds in the square lattice tensor network. $Z(\xi)$ is a constant with respect to $m$, to be chosen such that the limit

$$C(m) := \lim_{\xi \to \infty} C(m, \xi) \tag{49}$$

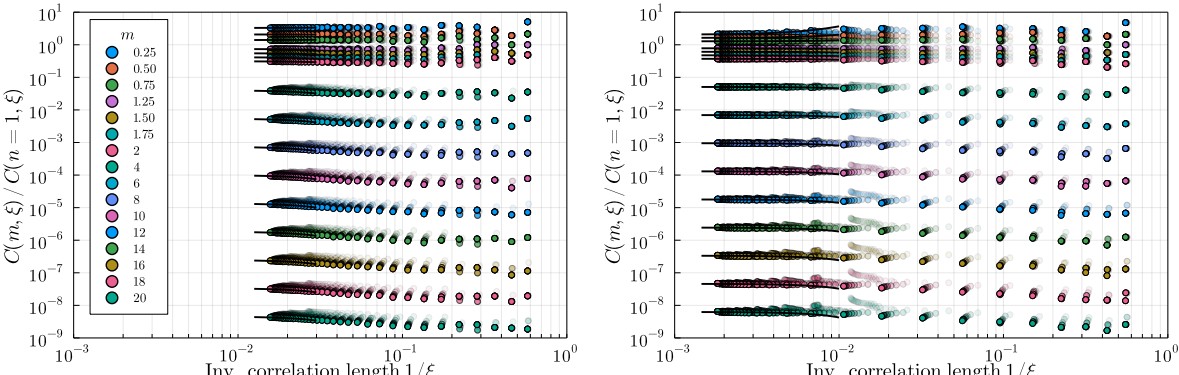

Figure 9: Collective analysis for the quantity $C(m, n = 1)$ at $\chi_B = 2$ and $\chi_B = 6$, for different values of $m$. A light to dark shading has been used for increasing environment bond dimensions $\chi_E^{\text{eff}}$. The fit to extract the infinite correlation length limit has been performed for $\chi_E^{\text{eff}} = 1000$ using Eq. (51). The legend applies to both panels.

is finite. Such a (renormalization) constant is guaranteed to exist, but it is quite intricate to compute it carefully, since $Z$ diverges itself in the proximity of critical points. While this is usually happening in a power-law fashion, in strict relation to what we usually call critical exponents (or anomalous dimensions), in the context of asymptotic freedom the divergence of $Z$ is itself logarithmically slow (a.k.a., the associated operator is RG-marginal). Therefore, a much cleaner approach is to define instead the ratio of such correlation values

$$\mathcal{C}(m, n) \coloneqq \lim_{\xi \to \infty} C(m, \xi)/C(n, \xi) = C(m)/C(n) , \tag{50}$$

thus circumventing the need to determine $Z(\xi)$ explicitly for different values of $\xi$. We will set $n = 1$ as the reference distance in the following.

In principle, one could first extrapolate the raw correlations to the limit of infinite bond dimension $(\chi_B, \chi_E^{\text{eff}} \to \infty)$ for each value of the coupling $\beta$, as we did in Section 4.3, and then define $C(m, n, \xi)$ to perform the $\xi \to \infty$ limit of Eq. (50). Alternatively, we could also consider first the raw correlation data $C(m, n, \xi(\chi_B, \chi_E^{\text{eff}}))$ as a function of $\xi(\chi_B, \chi_E^{\text{eff}})$, and then extrapolate to the limit of $\xi \to \infty$ – most practically, first for each $\chi_B$ separately. Both approaches lead to fully compatible results, though the second one turns out to be numerically more stable. In Fig. 9 we display the second approach for $\beta \leq 5$, which extends beyond the window of overlap with the mentioned Monte Carlo data of Ref. [84]. While the behaviour fluctuates strongly for small $\beta$ (which correspond to small $\xi$), it smoothens out for larger values, and an extrapolation is possible. Here we employ a simple fit of the form

$$C(m, n, \xi) = a + b\xi^{-2} . \tag{51}$$

Additional logarithmic corrections of the form $\text{poly}(1/\log(1/\xi))$ predicted by the Symanzik effective theory of lattice artefacts [88,89] should also be included, if one is interested in the precise quantitative value of the limit as the QCD community is [86,87]. This however goes beyond the scopes of this first tensor network analysis, considering the currently accessible bond dimensions. In Fig. 10 and Table 3 we show the extrapolated physical values $C(m, n = 1, 1/\xi \to 0)$ for different bond dimensions $\chi_B$ of the partition function approximation. While $\chi_B = 2$ is a too rough approximation, allowing for a

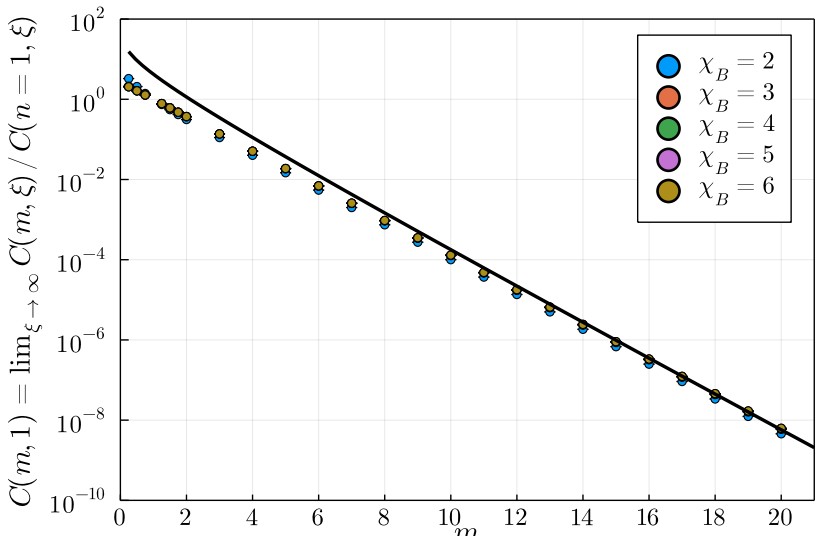

Figure 10: Infinite correlation length limit $\mathcal{C}(m,n) = \lim_{\xi \to \infty} C(m,\xi)/C(n,\xi)$ for all available bulk bond dimensions $\chi_B$ at different values of $m$. The reference is fixed to $n = 1$.

limited range of $1/\xi$ only, the differences for $\chi_B = 3$ up to $\chi_B = 6$ are hardly visible at this stage. This is however expected to change as soon as larger $\beta$ values are introduced in the analysis, and quantitative corrections to the extrapolated limits could be foreseen. They are however not forecasted to substantially change the behaviour in the logarithmic scale of such plot, as much as the error bars also do not (see Table 3), so we can safely leave them for future investigations. Most importantly, the behaviour of the (physical) correlation function at large distances is predicted by the so-called spectral representation

$$\mathcal{C}(m,1) = A\,F(m) + O\left(e^{-3m}\right) \equiv A \int \frac{\mathrm{d}p}{2\pi} \frac{\mathrm{e}^{-mE(p)}}{2E(p)} + O\left(e^{-3m}\right) \tag{52}$$

with $E(p) = \sqrt{p^2 + 1}$, which roughly gives $C(m,1) \simeq \exp(-m)/\sqrt{m}$. In Fig. 10, we display the result of a single-parameter fit to adjust the multiplicative constant $A$ to our data points for $m \geq 10$, which results in $A = 63.19 \pm 0.06$. The agreement is evidently already very good, and could possibly improve with future data sets for larger $(\chi_B, \chi_E^{\text{eff}})$, though this could be quite a challenging task.

Summarising, we seem to obtain consistency of the data computed via our tensor network approach with the predictions of asymptotic freedom, though strictly speaking we are not providing evidence that this limit $\xi \to \infty$ does take place indeed at $T \to 0$ and not earlier. This leaves the door open for alternative scenarios as, e.g., the lattice model having its critical coupling at a value larger than zero, as suggested by some mathematical physicists [90, 91].

## 5   Conclusions and outlook

In this work, we have presented a fresh approach for the study of the two-dimensional classical Heisenberg model based on tensor networks. The partition function for the statistical mechanics model is obtained by the contraction of a network of interconnected tensors residing on the lattice sites. The local

tensors are constructed from an analytic rewriting of the Boltzmann factors, whose precision is controlled by a cutoff in the infinite sum over the rotational group characters (i.e., angular momenta). The correct fusion of angular momenta at each lattice vertex is automatically ensured by directly imposing the $O(3)$ symmetry at the level of the constituent tensors.

Our technique gives direct access to thermodynamic quantities and physical observables for a formally infinite system over a wide (classical) temperature range. Noticeably, we can directly access a wealth of information about correlations in the system via the transfer matrix spectrum in our formalism, without the need for direct measures and/or for exponential fittings at prohibitively large distances. The finite bond dimensions employed throughout the algorithm play the effective role of spatial and thermal cutoffs. Therefore, data have to be extrapolated accordingly, which we did following protocols recently introduced in the (quantum) tensor network context. Moreover, we explored the behaviour of different entanglement measures, leveraging on the classical to quantum correspondence in different dimensionality.

Our findings bring new clues in the ever-lasting riddle about this fundamental model: On the one hand, the spin-spin correlation functions, once processed as commonly done in the lattice gauge theory community, display a satisfying agreement with the vastly endorsed mechanism of asymptotic freedom. This predicts a very rapidly diverging correlation length at low-temperatures but no finite temperature transition of any kind, i.e., the spin system remaining disordered down to zero temperature and the corresponding $O(3)$ non-linear sigma model remaining gapped down to zero coupling. The so-called physical value of the correlations, i.e., the one extrapolated in the zero lattice spacing – or, equivalently, infinite correlation length – limit, indeed matches the behaviour forecasted via the spectral representation.

On the other hand, the (largest) correlation length itself shows at first a striking agreement with a fitting form usually regarded as a bona fide signature of a Berezinskii-Kosterlitz-Thouless transition at a finite temperature $T_c \simeq 0.5J$. We incidentally stress here that the above mentioned extrapolation to the physical limit – as it is performed – does not strictly imply that this is reached at zero temperature. Our numerical results, however, suffer from severe approximations in the very-low temperature limit and important deviations appear close to such a putative $T_c$. We also cannot exclude the onset of some quasi-critical behaviour at very low temperatures. This seems in turn to be further supported by the entanglement-based quantities we have been looking at, for which we observe a strong entanglement-scaling dependence in the low-temperature regime.

We could envision future (larger-effort) investigations to compute further quantities like Noether-currents correlations, form factors as well as others and compare their behaviour with the exactly computable ones for the integrable $O(3)$-NLSM, in order to let the asymptotic freedom mechanism undergo more stringent tests. Conversely, the availability of reliable data for lower and lower temperatures shall also contribute to lift the mystery about the apparent finite-$T$ BKT-transition and/or the quasi-critical behaviour. Furthermore, even more ingenious extrapolation schemes from the achievable finite bond dimension data would be of the utmost help in elucidating the subtle details of this intriguing problem. An example could consist in quantifying the intrinsic cutoff of the characters' expansion with a single parameter, and then searching for some sort of regression plane to achieve the simultaneous limit of both truncations of our ansatz.

Put into a broader perspective, the method that we have employed here offers a strong potential for the detailed and systematic exploration of a wealth of classical and quantum spin models. It will be interesting to flesh out its applications for physical systems defined on frustrated geometries and for real materials where tensor networks have shown great promises recently [47, 92–97]. These efforts will help us in gaining further insights into exotic phases of matter such as quantum spin liquids including their finite temperature properties and quantum to classical crossovers, a topic of intense

current interest.

# Acknowledgements

We would like to thank for insightful discussions with Karin Everschor-Sitte, Andreas Haller, Haye Hinrichsen, Agostino Patella, Peter van Dongen, and Yizhi You. We acknowledge computation time at the HPC clusters Curta, Freie Universität Berlin, and JURECA, Forschungszentrum Jülich.

During the preparation of the manuscript we became aware of a similar study of the Heisenberg and related $RP^2$ models by Lander Burgelman, Laurens Vanderstraeten and Frank Verstraete [98]. We acknowledge a stimulating exchange and insightful suggestions to improve the finite-entanglement scaling.

**Funding information.** We acknowledge DFG funding through CRC 183 (B01), for which this is an inter-node Berlin-Cologne project of the CRC, EI 519/15-1, GZ OR 381/3-1 as well as GZ SCHM 2511/10-1. This work has also received funding from the Cluster of Excellence MATH+ and the European Union's Horizon 2020 research and innovation programme under grant agreement No. 817482 (PASQuanS). R. O. also acknowledges support from Ikerbasque and the DIPC.

# A    Numerical data for the consistency check with asymptotic freedom

For completeness and reproducibility we show the final infinite correlation length limit $\mathcal{C}(m,n) = \lim_{\xi\to\infty} C(m,\xi)/C(n,\xi)$ used in the analysis of Section 4.4 in Table 3.

| $m$ | $\chi_B$ | | | | |
| --- | --- | --- | --- | --- | --- |
| | 2 | 3 | 4 | 5 | 6 |
| 0.25 | 3.284(31) | 2.116(18) | 2.062(6) | 2.054(6) | 2.050(4) |
| 0.50 | 2.078(21) | 1.634(3) | 1.629(1) | 1.627(1) | 1.626(1) |
| 0.75 | 1.395(10) | 1.279(1) | 1.279(1) | 1.278() | 1.278() |
| 1.25 | 7.422(56)e$-$1 | 7.812(2)e$-$1 | 7.807(2)e$-$1 | 7.807(2)e$-$1 | 7.813(2)e$-$1 |
| 1.50 | 5.515(32)e$-$1 | 6.096(3)e$-$1 | 6.091(2)e$-$1 | 6.096(2)e$-$1 | 6.099(2)e$-$1 |
| 1.75 | 4.159(26)e$-$1 | 4.754(2)e$-$1 | 4.747(1)e$-$1 | 4.755(2)e$-$1 | 4.758(2)e$-$1 |
| 2.00 | 3.126(19)e$-$1 | 3.705(2)e$-$1 | 3.701(1)e$-$1 | 3.707(1)e$-$1 | 3.712(1)e$-$1 |
| 3.00 | 1.107(6)e$-$1 | 1.367(1)e$-$1 | 1.369()e$-$1 | 1.371(1)e$-$1 | 1.374(1)e$-$1 |
| 4.00 | 3.999(28)e$-$2 | 5.041(2)e$-$2 | 5.056(3)e$-$2 | 5.072(5)e$-$2 | 5.083(5)e$-$2 |
| 5.00 | 1.470(10)e$-$2 | 1.860(1)e$-$2 | 1.869(1)e$-$2 | 1.876(2)e$-$2 | 1.882(3)e$-$2 |
| 6.00 | 5.423(42)e$-$3 | 6.854(3)e$-$3 | 6.902(7)e$-$3 | 6.938(11)e$-$3 | 6.965(14)e$-$3 |
| 7.00 | 1.993(15)e$-$3 | 2.529(1)e$-$3 | 2.550(3)e$-$3 | 2.567(5)e$-$3 | 2.577(6)e$-$3 |
| 8.00 | 7.401(55)e$-$4 | 9.323(5)e$-$4 | 9.425(15)e$-$4 | 9.492(22)e$-$4 | 9.544(26)e$-$4 |
| 9.00 | 2.722(16)e$-$4 | 3.439(2)e$-$4 | 3.482(6)e$-$4 | 3.511(9)e$-$4 | 3.532(11)e$-$4 |
| 10.00 | 9.993(61)e$-$5 | 1.268(1)e$-$4 | 1.286(3)e$-$4 | 1.299(4)e$-$4 | 1.308(5)e$-$4 |
| 11.00 | 3.679(30)e$-$5 | 4.676(3)e$-$5 | 4.755(11)e$-$5 | 4.803(17)e$-$5 | 4.839(20)e$-$5 |
| 12.00 | 1.355(11)e$-$5 | 1.725(1)e$-$5 | 1.757(4)e$-$5 | 1.777(7)e$-$5 | 1.791(8)e$-$5 |
| 13.00 | 4.979(34)e$-$6 | 6.356(6)e$-$6 | 6.490(18)e$-$6 | 6.571(27)e$-$6 | 6.633(32)e$-$6 |
| 14.00 | 1.836(13)e$-$6 | 2.344(3)e$-$6 | 2.398(7)e$-$6 | 2.430(11)e$-$6 | 2.454(13)e$-$6 |
| 15.00 | 6.772(52)e$-$7 | 8.645(10)e$-$7 | 8.864(29)e$-$7 | 8.990(44)e$-$7 | 9.084(52)e$-$7 |
| 16.00 | 2.483(18)e$-$7 | 3.189(4)e$-$7 | 3.274(12)e$-$7 | 3.325(18)e$-$7 | 3.363(21)e$-$7 |
| 17.00 | 9.182(69)e$-$8 | 1.176(2)e$-$7 | 1.210(5)e$-$7 | 1.230(7)e$-$7 | 1.245(8)e$-$7 |
| 18.00 | 3.357(28)e$-$8 | 4.336(7)e$-$8 | 4.470(18)e$-$8 | 4.549(27)e$-$8 | 4.605(32)e$-$8 |
| 19.00 | 1.236(9)e$-$8 | 1.599(3)e$-$8 | 1.650(7)e$-$8 | 1.682(11)e$-$8 | 1.705(13)e$-$8 |
| 20.00 | 4.559(35)e$-$9 | 5.897(10)e$-$9 | 6.101(29)e$-$9 | 6.223(41)e$-$9 | 6.311(49)e$-$9 |

Table 3: Numerical data for the infinite correlation length limit of $\mathcal{C}(m,n) = \lim_{\xi \to \infty} C(m,\xi)/C(n,\xi)$, for all available bulk bond dimensions $\chi_B$ at different values of $m$. As in the corresponding Fig. 10, the reference is set to $n = 1$.

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
