# Peer review of "The classical two-dimensional Heisenberg model revisited: An $SU(2)$-symmetric tensor network study"

_SciPost Physics_

## Round 2 · Referee Report · Anonymous (Referee 1) · 2021-8-12

Report
In this manuscript, the authors reported a large-scale tensor network numerical study of the two-dimensional (2D) classical Heisenberg model with O(3) symmetry.
The classical Heisenberg model is a paradigmatic model in statistical physics and condensed matter. While extensive analytical and numerical efforts were devoted to this model in the past, it remains a controversial issue whether the 2D classical O(3) Heisenberg model on the square lattice exhibits a Berezinskii-Kosterlitz-Thouless (BKT) phase transition at finite temperature.
Taking advantage of their tensor network library incorporating SU(2) symmetry, the authors address this important open problem. After casting the partition function into a tensor network, a symmetry-preserving corner transfer matrix renormalization group (CTMRG) approach is employed to (approximately) contract the tensor network in the thermodynamic limit. Within this framework, various physical quantities, including spin-spin correlation functions, thermodynamic entropy, and entanglement measures, have been computed. In particular, a rapidly diverging correlation length is observed at low temperature, which is consistent with the two contradictory hypotheses in the literature: finite-temperature BKT transition and asymptotic freedom (i.e., absence of finite-temperature transition).
With decreasing temperatures, the truncation of angular momentum basis in the partition function causes increasingly large errors. However, this is unavoidable for carrying out CTMRG contractions. It is then extremely difficult to conclusively answer whether the transition temperature is exactly zero or very low. Thus, this leaves room for future investigations.
In my opinion, the results reported in this manuscript surely represent state-of-the-art tensor network calculations and certainly shed new light on an important open problem. The manuscript is very well written. It would also motivate further development of tensor network methods and applying them to other paradigmatic models. Therefore, I would warmly recommend its publication in SciPost.
Below I have two minor comments:
1) In page 8, the claim that “the model is Bethe integrable” is a bit confusing. As far as I know, the computation of the partition function for the one-dimensional classical Heisenberg model relies on (directly) diagonalizing the transfer matrix in the angular momentum basis, so the Bethe ansatz technique is not used here, nor in Refs. [71,72].
2) In page 9, the tensor network representation of the classical O(3) Heisenberg model reminds me of an earlier paper [Liu et al., Phys. Rev. D 88, 056005 (2013)], where a similar tensor network representation was derived in Sec. III B. If they coincide, it might make sense to add a reference.
Author: Philipp Schmoll on 2021-10-11 [id 1834]
(in reply to Report 1 on 2021-08-12)
Thank you very much for your careful reading, informative summary and positive comments. We appreciate your assessment of our results and the recommendation for publication.
Below we provide responses to the comments point-by-point.
1) We apologize for the confusion, the sentence is unfortunately fully misplaced. It should have appeared in the context of the 1D quantum Heisenberg model (and the associated 1D O(3) NLSM), which are Bethe integrable. 2) Thank you for pointing us to the reference. It is indeed of relevance to our study since it is an equivalent re-formulation of the Boltzmann factors.
We have addressed your feedback in a revised version of the manuscript, with the following changes.
1) We have moved the sentence regarding Bethe-integrability to the corresponding passage in the conclusion. 2) The reference [Liu et al., Phys. Rev. D 88, 056005 (2013)] has been added to the Bibliography.
Author: Philipp Schmoll on 2021-10-11 [id 1833]
(in reply to Report 2 on 2021-09-09)Thank you very much for your careful reading, informative summary and positive comments. We appreciate your assessment of our results and the recommendation for publication.

---

## Round 2 · Referee Report · Anonymous (Referee 2) · 2021-9-9

Report

---

## Editorial Decision

unknown